# Remodeling of the postsynaptic proteome in male mice and marmosets during synapse development

Takeshi Kaizuka[1,2,9], Takehiro Suzuki [3,9], Noriyuki Kishi[1], Kota Tamada [1,2], Manfred W. Kilimann [4], Takehiko Ueyama [5], Masahiko Watanabe[6], Tomomi Shimogori[1], Hideyuki Okano [1,7], Naoshi Dohmae [3] & Toru Takumi [1,2,8] ✉

Postsynaptic proteins play crucial roles in synaptic function and plasticity. During brain development, alterations in synaptic number, shape, and stability occur, known as synapse maturation. However, the postsynaptic protein composition changes during development are not fully understood. Here, we show the trajectory of the postsynaptic proteome in developing male mice and common marmosets. Proteomic analysis of mice at 2, 3, 6, and 12 weeks of age shows that proteins involved in synaptogenesis are differentially expressed during this period. Analysis of published transcriptome datasets shows that the changes in postsynaptic protein composition in the mouse brain after 2 weeks of age correlate with gene expression changes. Proteomic analysis of marmosets at 0, 2, 3, 6, and 24 months of age show that the changes in the marmoset brain can be categorized into two parts: the first 2 months and after that. The changes observed in the first 2 months are similar to those in the mouse brain between 2 and 12 weeks of age. The changes observed in marmoset after 2 months old include differential expression of synaptogenesis-related molecules, which hardly overlap with that in mice. Our results provide a comprehensive proteomic resource that underlies developmental synapse maturation in rodents and primates.

In the central nervous system, neurons communicate through synapses. The postsynaptic parts of excitatory synapses are formed mainly on small protrusions on the dendrites called dendritic spines[1,2]. The formation of neuronal circuits occurs throughout the developmental period, and maturation in dendritic spines is a crucial part of this process. During development, the number, structure, and dynamics of dendritic spines undergo significant changes[2]. In the mouse brain, the number of dendritic spines increases rapidly in the postnatal period and reaches its peak at the early juvenile stage (3 week-old)[3–5]. In the brain of primates, including humans, the number of dendritic spines or synapses observed with electron microscopy increases around perinatal and early postnatal periods and reaches its peak at the early juvenile stage[6–9]. The subsequent decrease of dendritic spines at the late developmental stage is known as 'pruning', which continues until

[1]RIKEN Brain Science Institute, Wako, Saitama 351-0198, Japan. [2]Department Physiology and Cell Biology, Kobe University School of Medicine, Chuo, Kobe 650-0117, Japan. [3]Biomolecular Characterization Unit, RIKEN Center for Sustainable Resource Science, Wako, Saitama 351-0198, Japan. [4]Max Planck Institute for Experimental Medicine, Göttingen 37075, Germany. [5]Laboratory of Molecular Pharmacology, Biosignal Research Center, Kobe University, Nada, Kobe 657-8501, Japan. [6]Department of Anatomy, Faculty of Medicine, Hokkaido University, Kita, Sapporo 060-8638, Japan. [7]Department of Physiology, Keio University School of Medicine, Shinjuku, Tokyo 160-8585, Japan. [8]RIKEN Center for Biosystems Dynamics Research, Chuo, Kobe 650-0047, Japan. [9]These authors contributed equally: Takeshi Kaizuka, Takehiro Suzuki. ✉e-mail: takumit@med.kobe-u.ac.jp

adulthood[2]. Spines arise from thin protrusions on dendrites, called filopodia[10,11]. Filopodia grow into spines by enlargement of the heads. Thus, thin spines are immature spines, whereas mushroom-shaped spines are considered mature spines. In the mouse cerebral cortex, the ratio of mushroom spines increases from about 40% to 80% between 2 week-old and 4 week-old[12]. Immature spines are highly dynamic and have a shorter lifetime than mature ones[11]. The turnover rate of spines in the mouse cortex shows about a 90% decrease from 2 week-old to 8 week-old[4].

Dendritic spines have been implicated in neuropsychiatric disorders. Data on synaptic molecules related to autism spectrum disorder (ASD) has been accumulating since neuroligin-3, a synaptic molecule, was first reported as a risk gene for ASD, and the concept of "synaptopathy" is currently used to broadly describe certain features of psychiatric or neurological diseases[13–15]. Alteration in spine density or morphology has been reported in patients and model animals of ASD, schizophrenia (SCZ), and Alzheimer's disease (AD)[2].

Synaptic plasticity also changes during development[16,17]. These changes are essential for strengthening or weakening synaptic connectivity and the formation and elimination of synapses. Proteins expressed on synapses play crucial roles in these processes, and most of these proteins are localized on the postsynaptic density (PSD).

The PSD is a protein complex beneath the plasma membrane of dendritic spines. The PSD is composed of more than 1000 proteins, including neurotransmitter receptors, cell-cell adhesion molecules, scaffolding proteins, and signaling enzymes[18,19]. The combination of these proteins makes spines functional synaptic compartments. The PSD can be biochemically purified using differential and density gradient centrifugation[20]. Proteome analysis of the PSD has been performed for various species under both physiological and pathological conditions[18]. During development, the composition of PSD also changes[21,22]. The alteration of PSD composition may be involved in synaptic maturation. So far, several studies have shown the developmental trajectory of gene expression and protein expression in rodents and primates[23–31]. In addition, studies focused on specific proteins in the PSD showed differences in the composition and assembly of protein complexes in the mouse brain at distinct developmental stages[32,33]. By contrast, relatively few papers have addressed alterations of PSD composition during the developmental period using a comprehensive approach. Proteome analysis of synaptic membrane showed alterations in synaptic protein composition in mouse visual cortex at postnatal day (P) 34-78 and rat medial prefrontal cortex at P30-46[28,29]. However, changes in PSD protein composition including younger periods (i.e. around P14-P21), where differences in number, shape, and turnover rate of spines are observed at a higher rate, have not been analyzed using an unbiased approach. Moreover, the developmental alteration of PSD composition in the primate brain has not been reported.

In this study, we performed a quantitative proteome analysis of the PSD of mouse brains at four time points and marmoset brains at five time points during postnatal development. We found that proteins related to synapse regulation are enriched in proteins differentially expressed in mice during this period. Systematic bioinformatics analyses using reported transcriptome datasets revealed a positive correlation between the relative abundance of mRNA and proteins in PSD, suggesting that transcriptional regulation is involved in PSD remodeling. Referring to transcriptome datasets of primate brains, we found that the decrease or increase of mRNA in mouse brains during postnatal development occurs around the perinatal period in humans and macaques. Proteome analysis of marmoset PSD during postnatal development described the different trajectories of PSD composition between mouse and marmoset. Comparing the developmental gene expression changes in humans (prenatal vs postnatal) with the transcriptome of ASD patient brains (control vs ASD), we found that developmental PSD remodeling appears to be defective in patients with ASD. Our results uncovered developmental trajectories of PSD

proteome in rodents and primates, which would be involved in the maturation of synapses.

## Results

### Proteome analysis of mouse PSD during postnatal development

We prepared brain samples from 2, 3, 6, and 12 week-old mice. Purification of PSD from brain tissue was performed using differential and sucrose density gradient centrifugation as reported previously[20] (See also Methods) (Supplementary Fig. 1a). We confirmed that a core scaffolding molecule in PSD, PSD-95, was enriched in the PSD fraction, whereas a presynaptic protein, synaptophysin, was mostly eliminated (Supplementary Fig. 1b). The prepared PSD samples were subjected to LC-MS/MS analysis. The experiment was performed four times independently. The resulting data were subjected to label free quantification (LFQ). In total, 4144 proteins were detected from 16 PSD samples. Among them, we found 270 stage-specific proteins (proteins detected in 4 of 4 replicates and 0 of 4 replicates in at least one stage, respectively) (Supplementary Data 1). About 80% of them were detected at 2 week-old and became undetectable before 12 week-old (Supplementary Fig. 2a). To analyze major synaptic components across development, we extracted proteins according to the following three criteria: (1) at least two unique peptides were identified, (2) quantified in all 16 datasets, and (3) coefficient of variation <100 in all ages. In the case when multiple proteins are encoded by a single gene, we selected a single protein encoded by a single gene whose signal intensity is highest. The result of this selection was 2186 proteins (Supplementary Data 2). Principal component analysis (PCA) of relative protein abundance showed that protein composition from 2 week-old to 12 week-old gradually changed (Fig. 1a). The data of 4 timepoints were distinguished in the first principal component (PC1), although the difference between 6 week-old and 12 week-old is relatively small. Heatmap analysis of individual protein abundance showed the subsets of proteins that gradually increased or decreased in abundance as development progressed (Fig. 1b). These data describe gradual transition of PSD composition from juvenile to adult stage.

### Distinct profile of the co-regulated protein clusters

To identify clusters of co-regulated groups of postsynaptic proteins, we performed k-means clustering. The three clusters were selected based on the best combination of indices provided by the NbClust R package (see Methods for details). The resulting Clusters 1, 2, and 3 were composed of 393, 1039, and 754 proteins, respectively (Fig. 1c, Supplementary Fig. 3, and Supplementary Data 2). The abundance of proteins in Cluster 1 gradually decreased but increased in Cluster 3, whereas the abundance of Cluster 2 proteins remained almost unchanged across development (Fig. 1c). We checked whether the proteins in each cluster include proteins reported to localize on PSD using (1) Gene Ontology (GO) analysis and (2) evaluation of the overlap with the proteome of PSD fraction reported in 16 datasets[34–48]. As expected, we found significant enrichment of the proteins known to be expressed on PSD (GOTERM_CC postsynaptic density) in the 2186 proteins, and >90% of the 2186 proteins were reported in at least 1 dataset (Fig. 1d, e). Although the enrichment and overlap were also found in each Cluster, the Cluster 1 proteins showed a relatively low enrichment and overlap rate (Fig. 1d, e). One possible reason for this result is that the 16 datasets referred to here are proteins detected in adult brains. To test this, we referred to the overlap of the Cluster 1-3 proteins with 512 proteins detected in the crude PSD fraction of P9 mouse[49]. We found that 374 (73%) of the proteins in the P9 PSD proteome overlap with the Cluster 1-3 proteins. They consist of 16.8%, 19.7%, and 13.6% of Cluster 1, 2, and 3 proteins, respectively, suggesting that Cluster 1 includes proteins expressed on PSD in the juvenile brain, as well as Cluster 2 and Cluster 3. To understand what kinds of proteins are included in each cluster, we performed GO analysis and pathway analysis using Metascape (Fig. 1f) and SynGO (Supplementary

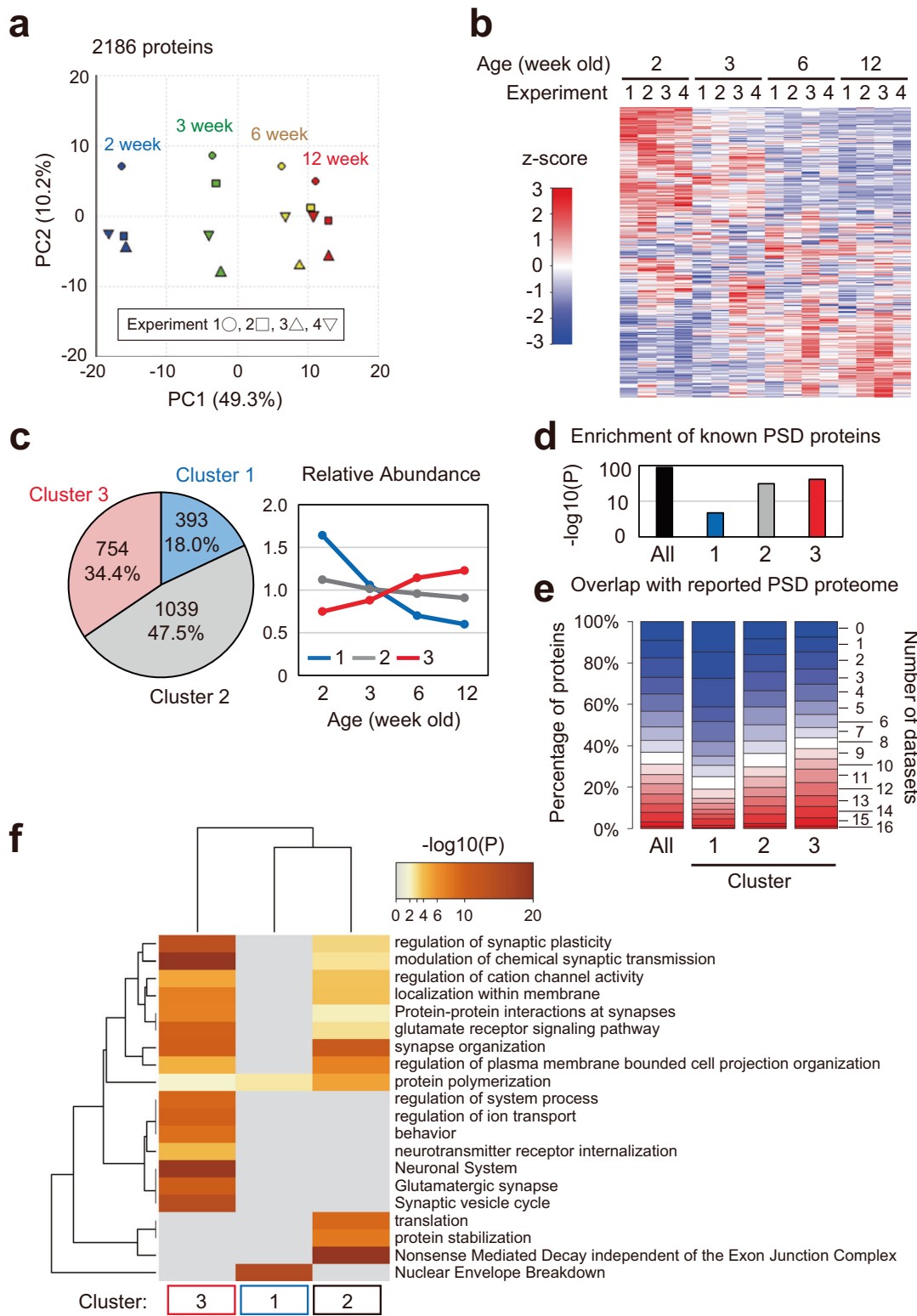

Fig. 4)[50,51]. We found that proteins expressed on synapse ("synapse organization" and "Protein-Protein interaction within synapse") are enriched in Clusters 2 and 3 (Fig. 1f). In addition, Cluster 3 showed high enrichment of proteins involved in modulation of synaptic function, such as "regulation of synaptic plasticity", "modulation of chemical synapse transmission", and "neurotransmitter internalization" (Fig. 1f). These proteins may contribute to the reorganization of neural circuits based on synaptic plasticity[16,17]. Considering the term "behavior", Cluster 3 proteins may also be involved in alteration of behavioral properties, such as increased activity during P28-42 and decreased fearful behavior during P24-75[52]. Although the result of Metascape does not provide much insight into the function of Cluster 1 proteins, analysis using SynGO showed that proteins involved in synaptic signaling are enriched in Cluster 1, as well as Cluster 2 and 3

**Fig. 1 | Remodeling of the postsynaptic density (PSD) during postnatal development in mice. a, b** PSD samples prepared from 2-, 3-, 6-, and 12 week-old mouse brains were subjected to LC-MS/MS to perform label-free quantification. Principal component analysis (PCA) was performed using the relative abundance values of 2186 proteins (**a**). The heatmap shows the relative abundance of each protein (**b**). **c** Identification of PSD protein clusters that are co-regulated during development using k-means clustering based on the mean protein abundance. (left) The pie chart shows the number of proteins in each identified cluster. (right) Expression profile (average of relative abundance) of the proteins in each cluster. **d** Enrichment of PSD proteins (GOTERM_CC postsynaptic density) in the 2186 proteins or proteins in each cluster described as raw *P*-values. **e** Overlap between the 2186 proteins or proteins in each cluster and 16 previously published PSD proteome datasets. **f** Statistically enriched Gene Ontology terms and pathway terms in each cluster identified by Metascape. Top20 terms are displayed as a hierarchically clustered heatmap. The heatmap cells are colored by their raw *P*-values; gray cells indicate the lack of enrichment for that term in the corresponding protein list. Source data are provided as a Source Data file.

(Supplementary Fig. 4). Signaling molecules in Cluster 1 include glycogen synthase kinase-3 beta (GSK3B), FERM, ARH/RhoGEF and pleckstrin domain protein 1 (FARP1), mitogen-activated protein kinase 8 (MAPK8), and phospholipase C beta 1 (PLCβ1; PLCB1) (Supplementary Data 2). We also analyzed the 270 stage-specific proteins. Two hundred twenty-one proteins detected in 4 of 4 replicates at 2- and/or 3 week-old are grouped as young age-specific proteins (termed "Young"), and 40 proteins detected in 4 of 4 replicates at 6- and/or 12 week-old are grouped as adult-specific proteins (termed "Adult"). Proteins in these groups are poorly overlapped with the previously reported PSD datasets (Supplementary Fig. 2b). The particularly low overlap of "Young" group proteins with known PSD proteins suggests that they are proteins expressed on PSD only at a young age and overlooked so far. In the Metascape analysis, we found that proteins related to "maintenance of postsynaptic specialization structure" is enriched in the "Young" group (Supplementary Fig. 2c). Taken together, these results show the developmental trajectory of the PSD proteome, which is thought to contribute to synaptic maturation in the developing mouse brain.

## Proteins that regulate synaptic function and dynamics are enriched in differentially expressed proteins

To extract proteins whose expression levels are significantly altered, we used two criteria; (1) Benjamini–Hochberg corrected *P*-value of one-way analysis of variance (ANOVA) was <0.05, (2) fold change (ratio of highest and lowest mean abundance) was >1.5. The 690 proteins that met both criteria were defined as differentially expressed (DE) proteins (Fig. 2a, Supplementary Data 2). We termed DE proteins in Cluster 1 (288) and Cluster 3 (267) as "Developmentally decreased postsynaptic proteins (DDP)" and "Developmentally increased postsynaptic proteins (DIP)", respectively (Fig. 2b, Supplementary Fig. 3). We first asked about the classification of 46 major proteins expressed on or associated with PSD, including adhesion molecules, glutamate receptors, scaffolding proteins, and signaling enzymes (Fig. 2c)[19]. Although 42 proteins belong to Cluster 2 or 3, four proteins are found in a DDP group in Cluster 1. The decrease of DLG3 (also known as SAP-102) and SHANK2 is consistent with previous reports[21,22,53]. We found that key enzymes involved in spine enlargement and synapse stabilization are included in the DIP group, including Ca2 + /calmodulin-dependent protein kinase (CaM kinase) (CAMK2A), Protein kinase C (PKC) (PRKCA, PRKCB, PRKCG), and Kalirin-7 (KALRN)[54,55]. To identify signaling pathways affected by the DE proteins in PSD, we performed canonical pathway analysis using Ingenuity Pathway Analysis. We found that the most enriched pathway in DDP and DIP proteins is "Synaptogenesis Signaling Pathway" (Fig. 2d and Supplementary Data 3). Indeed, we found that at least 126 proteins (55 "Decrease" and 71 "Increase") have been reported to affect the density, morphology, turnover, transmission, and/or plasticity of synapses (Supplementary Data 4). Among them, 75 proteins (32 in DDP and 43 in DIP) have been reported to affect spine density (Supplementary Data 4). This suggests that differential expression of the proteins on PSD is involved in the alteration of synaptic properties across development. Importantly, at least 66 proteins are reported to affect synaptic phenotype by RNAi or overexpression of the protein, suggesting that decreased or increased expression of these proteins can affect synapses (Supplementary

Data 4). Another enriched pathway, "Signaling by Rho Family GTPases", seems important as Rho GTPases are involved in synapse regulation[2]. To test whether the result of the pathway analysis is physiologically relevant, we examined developmental alteration of Rho GTPase signaling. Rho family GTPase signaling plays an essential role in synaptogenesis and synaptic plasticity[2,56,57]. In the dendritic spine, three Rho Family GTPases, RhoA, Rac1, and Cdc42, regulate assembly, disassembly, and severing of actin filament through phosphorylation of cofilin[10,57,58] (Fig. 2e). As actin filament and cofilin are distributed at cytoplasm within the dendritic spine rather than on PSD, we evaluated phosphorylation of cofilin in synaptosome, an isolated synaptic terminal compartment[59]. We found that phosphorylation of cofilin is decreased upon development in the crude synaptosome fraction obtained from the cortex and cerebellum (Fig. 2f). This result suggests alteration of actin remodeling via cofilin during postnatal development, which could be involved in altered shape and stability of spines. Taken together, these results suggest that remodeling of the PSD composition across development plays important roles in the alteration of synaptic density, dynamics, morphology, and plasticity in the mouse brain.

## Correlation of developmental changes between PSD protein level and mRNA level

What is the upstream mechanism of PSD composition remodeling across development? A simple explanation would be transcriptional regulation. That is, a decrease or increase in mRNA levels during this period results in a concomitant change in protein abundance. Thus, we referred to two transcriptome datasets of mouse cortex in development[23,30]. Heatmap analysis showed the tendency of correlation between PSD protein level and mRNA level in both datasets; the levels of mRNAs encoding DDP proteins tend to be decreased during development, and mRNAs that encode DIP proteins tend to be increased during development (Fig. 3a). This correlation supports our idea that transcriptional regulation is involved in alteration of PSD protein abundance. We found, however, that the timing of the alteration of mRNA abundance is earlier than that of proteins on PSD. We evaluated the correlation between gene expression and protein expression using Spearman's rank correlation coefficient (ρ). The correlation between quantitative changes of mRNAs at P15-adult (in Dataset 1) or P14-adult (in Dataset 2) and PSD protein at P14-adult is relatively low (ρ = 0.25 for both datasets) (Supplementary Fig. 5a). By contrast, changes of mRNA abundance from an earlier stage to an adult show a clear positive correlation with PSD protein at P14-adult (Supplementary Fig. 5a). In both datasets, the highest correlation was found on changes of mRNA abundance from P4 to an adult (ρ = 0.53 for both datasets). We focused on the proteins whose changes correlate with mRNA in both datasets. This resulted in the extraction of 171 proteins in the DDP group and 178 proteins in the DIP group (Supplementary Fig. 3, 5c). We termed these subgroups as "DDP correlated with gene expression in mouse (DDP-M)" and "DIP-M", respectively (Supplementary Fig. 3, 5c). The plot of mRNA abundance of DDP-M and DIP-M shows that they are altered mainly during the early postnatal period before P15 (Fig. 3b). If the abundance of DDP-M proteins and DIP-M proteins is transcriptionally regulated, how can we interpret the time lag between changes in mRNA level and protein level? We thought that

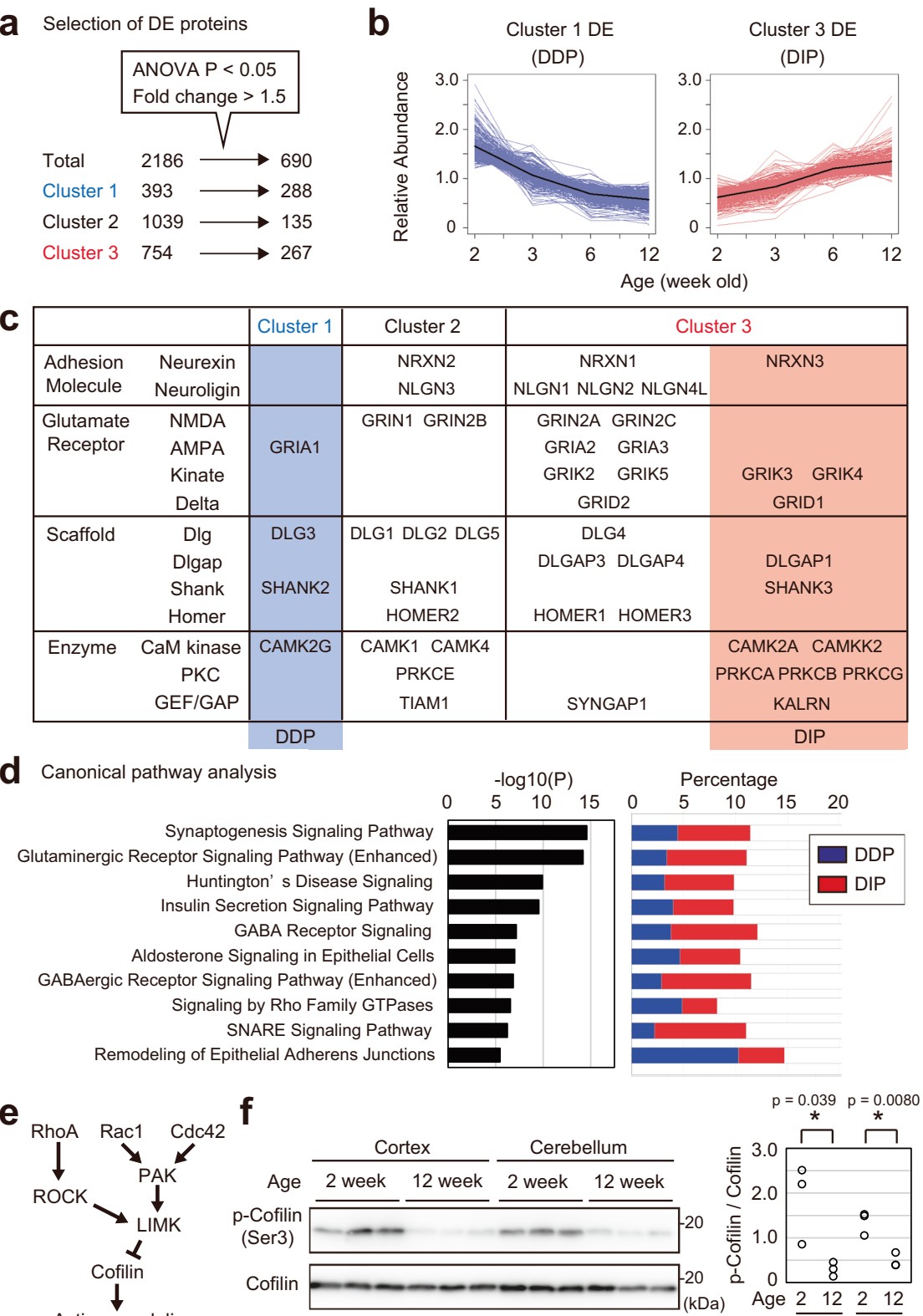

the time lag between alteration of mRNA abundance and protein abundance could be due to the slow protein turnover in neurons and referred to the dataset of protein half-life in mouse cortex[60]. The median half-life of the proteins in DDP-M and DIP-M was 8.32 days and 10.31 days, respectively (Fig. 3c). The long half-life may explain the time lag between the altered abundance of mRNA and protein, as simulated and experimentally described[61]. These data suggest that changes in gene expression levels are involved in the alteration of PSD composition during postnatal development of mice, and the alteration of protein abundance takes several days to become visible after mRNA levels change, as described in Fig. 3d. In addition to the changes in gene expression, we found changes in protein localization on PSD are also responsible for the transition of PSD composition. Immunoblotting analysis of the representatives of DDP (PLCβ1) and DIP (Paralemin-

**Fig. 2 | Proteins involved in synapse regulation are enriched in differentially expressed proteins. a** Selection of differentially expressed (DE) proteins using two criteria: (1) Benjamini–Hochberg corrected *P*-value of one-way analysis of variance (ANOVA) is <0.05, (2) max fold change is >1.5. Based on both criteria, 690 proteins were defined as DE proteins (See also Supplementary Data 2). **b** Expression profile of the DE proteins in Cluster 1 (termed 'DDP') and Cluster 3 (termed 'DIP'). Black lines indicate the average. **c** Classification of major proteins expressed on or associated with PSD. **d** Canonical pathway analysis using IPA. Pathways with the top 10 lowest raw *P*-values are shown. (left) Enrichment of indicated terms in DDP and

DIP described as *P*-value (right) Percentage of proteins included in each term. **e** Overview of Rho family GTPase signaling. Activation of any GTPase results in the inactivation of cofilin by phosphorylation. Cofilin affects the morphology of spines through actin remodeling. **f** Age-dependent reduction of cofilin phosphorylation. Crude synaptosome fractions were obtained from the cortex (CTX) or cerebellum (CB) of mice at indicated ages (*n* = 3 for each sample). 50 µg protein was loaded and phospho-cofilin and total cofilin were detected by immunoblotting. Quantification of band intensity ratio (p-Cofilin / Cofilin) is shown on the right. *\*P* < 0.05 (Two-sided unpaired Student's *t*-test). Source data are provided as a Source Data file.

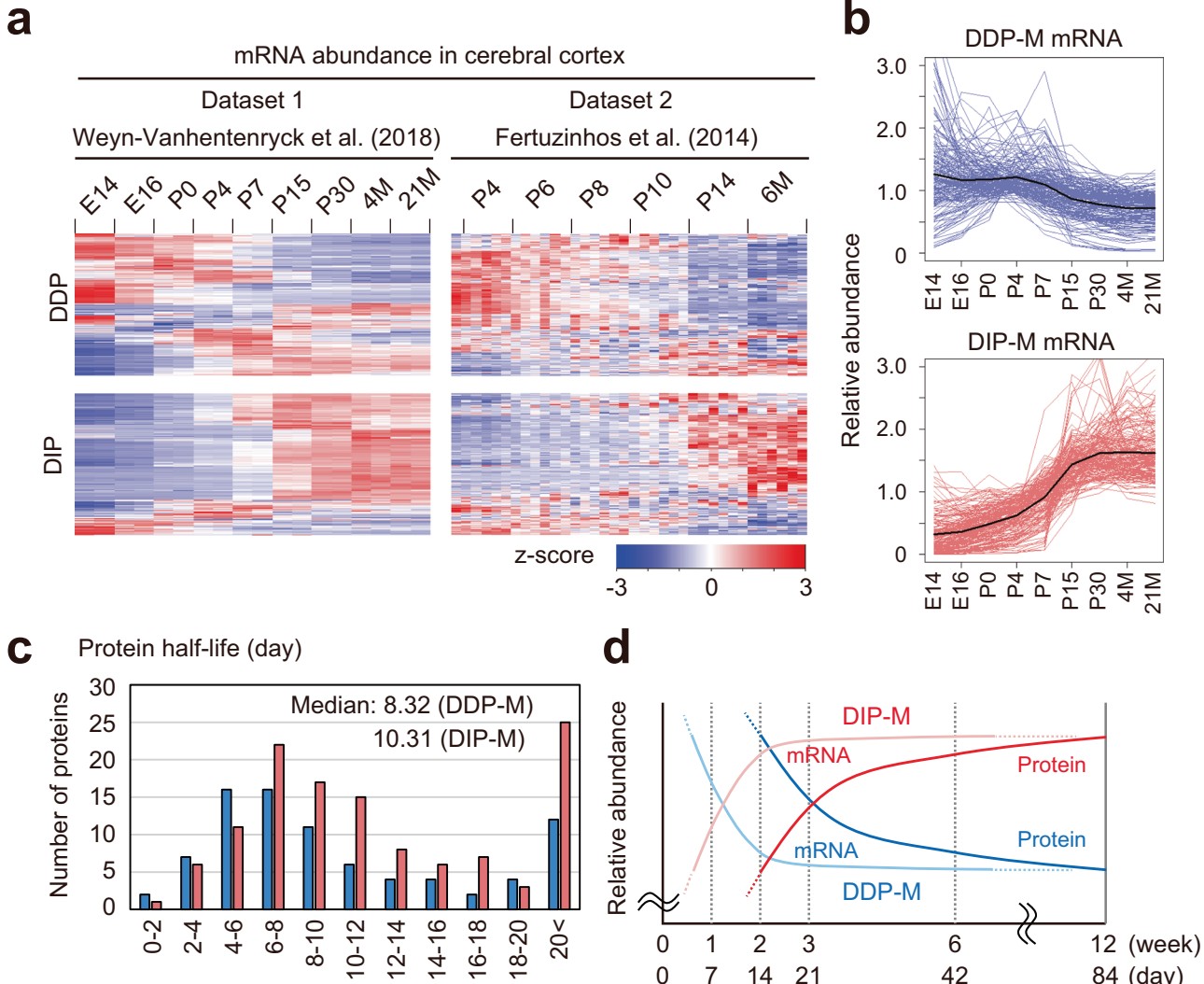

**Fig. 3 | Correlation of protein abundance on PSD and mRNA abundance in mouse cortex during postnatal development. a** Heatmap of the relative abundance of mRNAs that encode proteins in "Decrease" and "Increase" groups in the mouse cortex[23,30]. Six samples at each timepoint consist of 3 distinct cortical layers (infragranular layers, granular layer, and supragranular layers) from male and female mice. **b** Expression profiles of the mRNAs that encode proteins in the groups

DDP-M or DIP-M. Relative mRNA abundances (average of 6 samples) were plotted. Black lines indicate the average. **c** The plot of protein half-lives for the proteins in neurons[94]. **d** Model of the time course of mRNA abundance and protein abundance in the PSD. The changes in DDP or DIP of protein abundance in the PSD took place several days or weeks after the changes in mRNA abundance. Source data are provided as a Source Data file.

1; PALM) showed a significant decrease and increase of protein abundance in PSD during development without changes in total protein abundance (Supplementary Fig. 6).

## Switching of gene expression in primate brain at the perinatal period

Our next question is whether the PSD remodeling described above is also found in primates, including monkeys and humans. We analyzed

transcriptome datasets of developing neocortex of human and macaque[26,27]. As a result, we found similar changes in mRNA abundance between rodents and primates during development; mRNAs encoding the homologs of DDP-M and DIP-M proteins tend to be decreased and increased in the neocortex of developing humans and macaques, respectively (Fig. 4a and Supplementary Fig. 7a). Considering the correlation of mRNA and protein levels in the mouse brain, these data suggest that PSD remodeling found in mice also occurs in

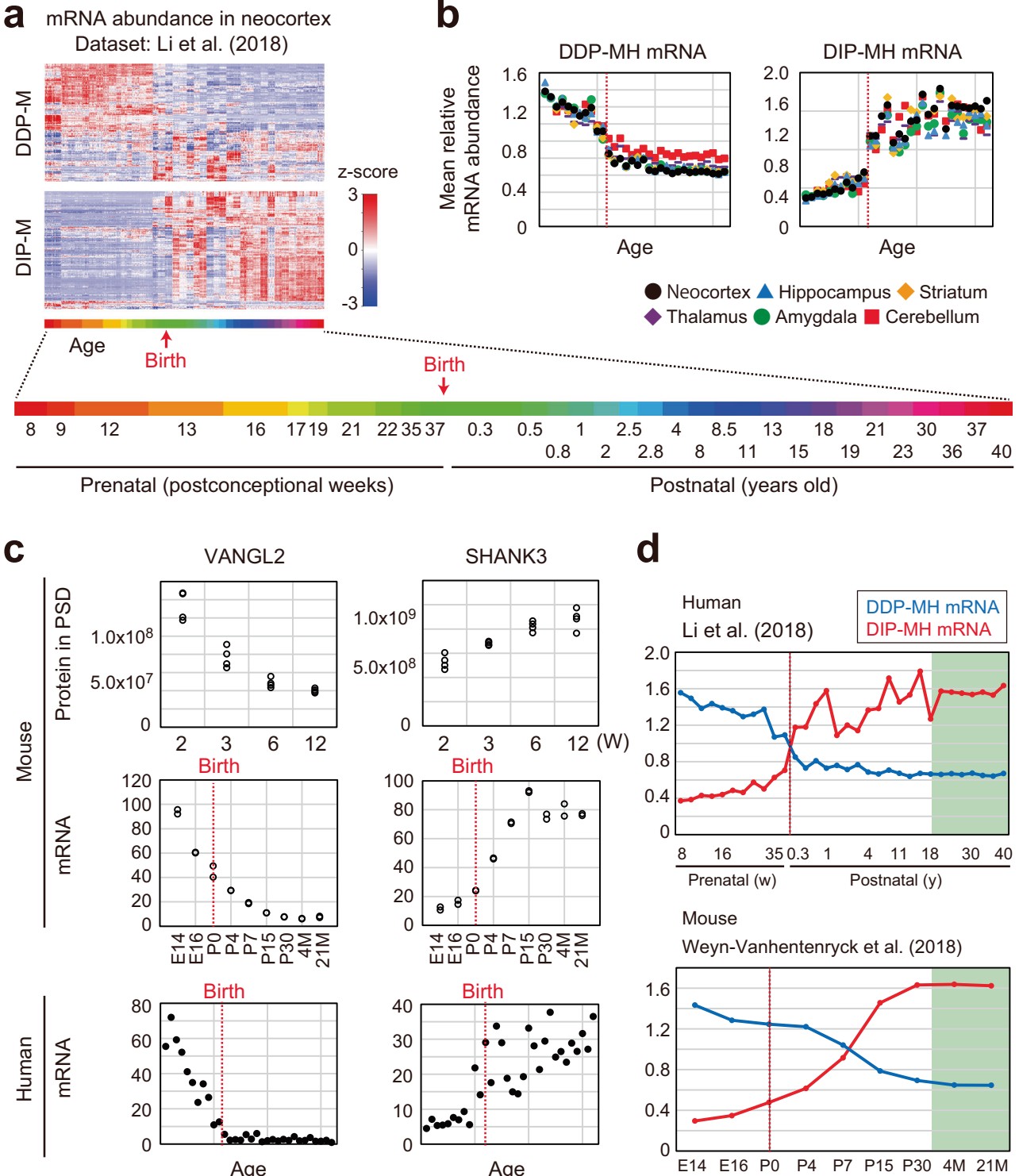

**Fig. 4 | Transcriptional profiles of genes encoding proteins on PSD in developing human brain. a** Heatmap of the relative abundance of mRNAs that encode DDP-M and DIP-M proteins in cortical regions in the developing human brain[26]. **b** The plot of the mean relative abundance of mRNAs that encode proteins in the groups DDP-MH (left) or DIP-MH (right). The 32-time points of the age of donors were described in (A). Red dotted lines indicate birth. **c** Representative examples of proteins in DDP-MH (left, VANGL2) and DIP-MH (right, SHANK3). (top) Protein abundance in the PSD obtained from mouse brains (Table S2). (middle) mRNA abundance in the mouse cortex[30]. (bottom) mRNA abundance in the human brain[26]. The 32 time points of the age of donors were described in (a). Red dotted lines indicate birth. **d** The time course of mRNA abundance of DIP-MH and DDP-MH proteins. Similar switching of gene expression occurs mainly in the perinatal period in primates and the early postnatal period in rodents. Source data are provided as a Source Data file.

the primate. It should be noted, however, that the alteration of mRNA abundance takes place around the perinatal period rather than the early juvenile period. This suggests that alteration of PSD protein composition occurs around the perinatal or early neonatal period in the primate brain, which is earlier than in mice. We termed the DDP-M and DIP-M proteins whose mRNA abundance is decreased and increased in the developing human neocortex as "DDP correlated with gene expression in mouse and human" (DDP-MH) (117 proteins) and DIP-MH (164 proteins), respectively (Supplementary Fig. 3). In the human neocortex, the mRNA abundance of DDP-MH and DIP-MH showed about a 2-3-fold change around the perinatal period (Fig. 4b). We also referred to the mRNA level of other brain regions: hippocampus, striatum, thalamus, amygdala, and cerebellum. The alteration of mRNA abundance was found in all of the regions in the human brain (Fig. 4b). Similarly, we extracted the DDP-M and DIP-M proteins whose mRNA abundance is decreased and increased in the developing macaque neocortex as "DDP correlated with gene expression in mouse and macaque" (DDP-MM) (142 proteins) and DIP-MM (159 proteins), respectively. Again, the alteration of mRNA abundance was found in all brain regions in the macaque (Supplementary Fig. 7b). These data suggest that the PSD remodeling takes place in various brain regions in the primate brain.

We show representative examples of DDP-MH and DIP-MH proteins in Fig. 4c and Supplementary Fig. 8. VANGL2 is a Wnt/planar cell polarity pathway component related to synapse formation[62], and SHANK3 is a scaffolding protein in PSD (Fig. 2c). mRNA of VANGL2 decreases before birth in humans, whereas this decrease occurs between prenatal and juvenile (P15) periods in mice. mRNA of SHANK3 was increased at 35–37 postconceptional weeks in humans, whereas it happened at the neonatal and early juvenile (P0-P15) period in mice. A similar expression pattern is also observed in the cases of HNRMPM and CAMK2A (Supplementary Fig. 8). To summarize, we plotted the developmental changes of mean relative abundance of mRNA in mice and humans in Fig. 4d. In the mouse brain, the changes in protein levels on PSD were further delayed and occurred between 2 week-old and 12 week-old (Figs. 3d, 4d). The different timing of gene expression changes is thought to be consistent with the timing of synaptogenesis in humans and mice[4,6]. We analyzed upstream transcriptional regulators of DDP-MH genes and DIP-MH genes referring to previously reported ChIP-Seq (chromatin immunoprecipitation sequencing) datasets using ChIP-Atlas[63] and analysis of transcription factor binding sites. As a result, we found enrichment of specific transcriptional regulators and transcription factor binding sites, suggesting that there are upstream regulators responsible for the developmental alteration of DDP-MH and DIP-MH (Supplementary Fig. 9a, b). In addition, we found a correlation between the expression of DDP-MH and DIP-MH genes and the epigenetic status of the genes (Supplementary Fig. 9c). These data suggest there are transcriptional regulators responsible for PSD remodeling (See also Discussion).

### Proteome analysis of PSD in the primate brain

Our analysis of transcriptome datasets suggested that alteration of PSD composition mediated by gene expression changes occurs during the perinatal period in the primate brain (Fig. 4 and Supplementary Fig. 7). To test this hypothesis and uncover overall changes of PSD composition in developing primates, we next performed proteome analysis of PSD in the primate. Considering the availability of samples, we opted to use common marmosets (*Callithrix jacchus*) instead of humans or macaques to obtain PSD from fresh primate brains. The common marmoset is a small New World primate that lives in stable extended families and has well-developed vocal communication[64]. Because of its similarity to humans, the marmoset is regarded as a useful model animal to study cognitive processes and mental illness[65]. We selected marmosets instead of macaques, considering the availability of fresh samples. To isolate PSD from limited brain samples, we

used a 3-step method (Supplementary Fig. 10a)[66] instead of a classical purification method. We confirmed the enrichment of PSD-95 and elimination of synaptophysin in a PSD fraction prepared with this method (Supplementary Fig. 10b). We found partial degradation of GluN2B (GRIN2B) and significant loss of PSD-95 in PSD fractions 8–24 h after death in the mouse brain, confirming the importance of fresh brain samples for PSD proteome analysis (Supplementary Fig. 11). Brain samples were then prepared from marmosets described in Supplementary Fig. 12a. We first analyzed PSD protein composition across seven brain regions using an adult (24-month-old) marmoset (Supplementary Fig. 12b and Supplementary Data 5). As a result, we detected a major difference between the forebrain and hindbrain, consistent with the observation in mice[37].

### Alteration of PSD protein composition during postnatal development in marmosets

We then analyzed the alteration of PSD composition during postnatal development in the marmoset. The neocortex and cerebellum were selected as representative regions of the forebrain and hindbrain, respectively. Considering the degree of cortical laminae maturation, level of serotonin, and expression of genes including serotonin receptor, P0 marmoset is assumed to be the equivalent stage of the mouse at 2 - 3 weeks old[67,68]. This suggests that alteration of PSD composition in mouse takes place in marmoset brain during the neonatal period, as well as human and macaque. Thus, PSD samples were prepared from 0 months (M), 2 M, 3 M, 6 M, and 24 M old marmosets. Two animals were used for each timepoint. Among the detected proteins, 3535 in the neocortex and 3845 in the cerebellum met the four criteria used for the mouse PSD proteome experiment described above (Supplementary Data 6, 7). PCA and heatmap analysis showed age-dependent alteration of PSD composition (Fig. 5a, b and Supplementary Fig. 13a, b). In both regions, the most significant difference was found between 0 M and 2 M. We asked whether the changes seen in the marmoset brain during the neonatal period were similar to that observed in the mouse brain after 2 week-old. To compare mouse and marmoset data, we acquired proteome data of mouse PSD using the experimental procedures used for marmoset. PSD samples were prepared from two biological replicates of the developing mouse cortex and cerebellum (at 2, 3, 6, or 12 weeks old) with the 3-step method. We found 3088 and 3586 proteins in the cortex and cerebellum, respectively (Supplementary Data 8, 9). Analysis of the protein overlap showed that the cortical and cerebellar PSD proteins found here include 1799 (82.3%) and 1746 (79.9%) proteins detected in whole-brain PSD proteome (Supplementary Fig. 14a). Similar to whole-brain PSD proteome, gradual changes in PSD composition after 2 week-old were observed both in the cortex and cerebellum (Supplementary Fig. 14b-e). The changes in PSD composition from 2 to 12 weeks old in whole-brain were positively correlated with that observed in cortical or cerebellar PSD prepared with the 3-step method ($\rho = 0.39$ in both regions). When we focused on DDP-MH and DIP-MH proteins, a more significant correlation was observed ($\rho = 0.66$ in the cortex and $\rho = 0.71$ in the cerebellum) (Supplementary Fig. 14f, g). We then compared these mouse proteome data with the marmoset proteome data described above. We detected 2531 (72.8%) proteins detected in marmoset neocortical PSD also in mouse cortical PSD, and 2778 (73.7%) in marmoset cerebellum PSD were detected in mouse cerebellum PSD (Supplementary Fig. 15a, b). Comparison of mouse and marmoset data showed that the PSD composition of the mouse at 2 week-old is relatively similar to that of the marmoset at 0 month-old (Fig. 5c and Supplementary Fig. 15c). In the cortex, the PSD composition of the mouse at 12 week-old showed high similarity to that of marmoset at 2- and 3 month-old but not 24 month-old, suggesting that primate brains undergo PSD remodeling that is not observed in mice at later developmental stages (Fig. 5c). Similar tendency is also observed in cerebellum, although PSD of 12 week-old mouse still keeps high similarity

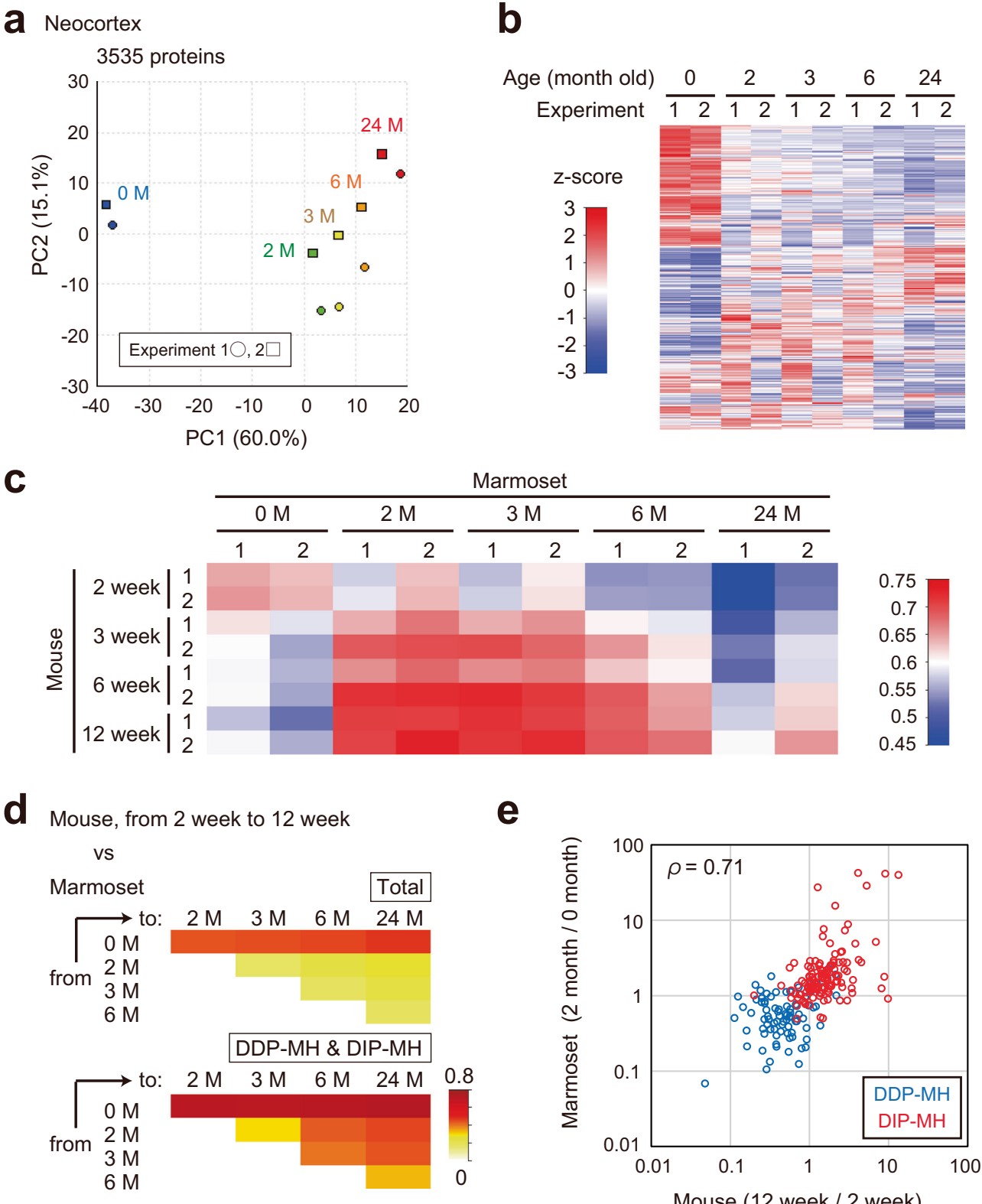

**Fig. 5 | Alteration of PSD proteome in marmoset neocortex during development. a** and **b** PSD samples prepared from 0-, 2-, 3-, 6-, and 24 month-old marmoset neocortices were subjected to LC-MS/MS for label-free quantification. PCA was performed using the relative abundance values of 3,535 proteins (**a**). The heatmap shows the relative abundance of each protein (**b**). **c** Pearson correlation coefficient of protein abundance (LFQ intensity) in PSD of mouse cortex and marmoset neocortex prepared with 3-step method. **d** Spearman's rank correlation coefficient of relative PSD protein abundance changes in mouse cortex and marmoset neocortex. For mouse data, the proteome of PSD prepared with the 3-step method was referred to. **e** Log10 fold change of the abundance of proteins in DDP-MH and DIP-MH groups in the developmental mouse cortex (12 week-old vs. 2 week-old) plotted against that of marmoset neocortex at neonatal period (2 month-old vs 0 month-old). Source data are provided as a Source Data file.

to 0 month-old marmoset in this region (Supplementary Fig. 15c). We found a correlation between changes of protein abundance in the mouse brain at 2–12 week-old and that in the marmoset brain after 0 M (Fig. 5d and Supplementary Fig. 15d). When we focused on DDP-MH and DIP-MH proteins, we observed more significant correlation ($\rho = 0.71$ in both regions) (Fig. 5d, e and Supplementary Fig. 15d, e). These results indicate that changes in PSD composition that occur in mouse brains after 2 week-old occur in marmoset brains after birth, primarily within the first 2 months.

### Alteration of PSD protein composition during late developmental stage in marmosets

Like other primate species, including humans, macaques, and chimpanzees, the marmoset brains undergo synaptic pruning during late postnatal development[6–9]. In the marmoset cortex, synapse density becomes highest around 3 M old and decreases after that[7] (Fig. 6a). Considering that alteration of synaptic protein composition can be involved in this process, we next focused on changes of PSD composition in the marmoset neocortex after 2 M. The changes after 2 M include a decrease of DDP-MH proteins and an increase of DIP-MH proteins continued from the 0–2 M period, which is shown as a positive correlation between changes in mice at 2–12 week-old and marmoset at 2–24 M (Figs. 5e and 6b). However, the overall changes observed in marmosets during 2–24 M were poorly correlated with those during 0–2 M (Fig. 6c), suggesting that changes in synaptic protein composition in marmoset brains in a late developmental stage are distinct from those that take place during the neonatal period. We considered a hypothetical model of developmental PSD remodeling as described in Fig. 6d. The decrease in DDP-MH proteins and increase in DIP-MH proteins in rodent brains during the juvenile to adult period occur in primate brains mainly during the neonatal period and continue until adulthood (Phase 1). Besides, there are distinct changes after a juvenile period in the primate brain (Phase 2). To clarify the changes in PSD composition in Phase 2, we next extracted proteins that showed significant changes. Given the difficulty in detecting statistical significance due to the limited sample size ($n = 2$), we focused proteins that showed >2-fold changes of average abundance between 2 M and 24 M. As a result, we found 504 decreased proteins and 142 increased proteins, which are termed Late DDP and Late DIP, respectively (Fig. 6e). Pathway analysis using IPA showed that proteins involved in "Synaptogenesis Signaling Pathway" are enriched in Late DDP and Late DIP, suggesting their involvement PSD composition in synapse remodeling at later developmental stages (Fig. 6f, Supplementary Data 3). There is some overlap of pathways between DDP/DIP and Late DDP/DIP, including Synaptogenesis Signaling Pathway, GABAergic Receptor Signaling Pathway (Enhanced), and Huntington's Disease Signaling (Figs. 2d and 6f). However, there is little overlap of proteins between DDP/DIP and Late DDP/DIP (Fig. 6g). There is also little overlap of proteins classified into Synaptogenesis Signaling Pathway, suggesting that distinct proteins related to synapse regulation are differentially expressed at Phase 1 and Phase 2 (Fig. 6h). Considering the enrichment of pathways related to GABAergic receptor, we questioned whether developmental trajectories of excitatory and inhibitory synapses differ between mice and marmosets. It is reported that the ratio of inhibitory neurons and synapses is higher in primate brains compared to rodents[69]. When we compared the protein abundance of major components of ionotropic glutamate receptor and GABA$_A$ receptor detected in both cortical datasets, we found that the abundance of GABA receptor subunits tends to be higher than that of NMDA and AMPA receptors in marmoset PSD (Supplementary Fig. 16). We also found that the abundance of GluN1 subunit and GluN2B subunit of NMDA receptor shows ~50% decrease from 2 M to 24 M in marmoset brain, whereas the abundance of GABA receptor α1 subunit is gradually increased and exceeds that of GluN1 and GluN2B. If the protein abundance reflects synapse number, this result

suggests a developmental increase of inhibitory synapses in primate brains. Taken together, our results uncovered trajectory of developmental marmoset PSD proteome that can be involved in synapse maturation process in primate brain.

### Dysregulation of genes encoding PSD proteins in patients with ASD

Lastly, we investigated the dysregulation of PSD composition in neuropsychiatric disorders. We found that proteins related to various diseases are enriched in our PSD proteome (Supplementary Fig. 17a). Notably, proteins increased during development show high enrichment of proteins related to neuropsychiatric disorders, including ASD (Supplementary Fig. 17b, c). ASD is a neurodevelopmental disorder that is characterized by impaired social interaction and communication and restricted and repetitive behaviors, activities, or interests[70,71]. Synaptic abnormalities are common phenotypes in individuals with ASD and ASD model mice[2,13,15,70,72]. In particular, immaturities of the dendritic spines, such as enhanced turnover, thinner shape, and defective pruning, have been observed across a wide range of ASD model animals and patients with ASD[2,4,5,73,74]. Thus, we investigated whether there was a defect in gene expression related to synapse maturation. We analyzed the transcriptome dataset of ASD patient brains[75]. We focused on genes differentially expressed in a cortical region of patients with ASD using $P < 0.05$ as a cutoff (P-value before adjusting for multiple testing). In this criterion, approximately 30% of genes showed differential expression in the neocortex of individuals with ASD (Supplementary Fig. 18a). It is reported that expression of the genes encoding synaptic proteins is affected in ASD patient brains[76,77]. Consistent with this, the genes encoding proteins that we detected on PSD (2186 proteins in Supplementary Data 2) are enriched in DE genes of patients with ASD (Supplementary Fig. 18a). DE genes were further enriched in the "DIP-MH" group, and 86% (62/72) were downregulated in individuals with ASD. Although DE genes were not enriched in the "DDP-MH" group, the proportion of upregulated genes was significantly higher compared to all PSD proteins (Supplementary Fig. 18a). The plot of mRNA level alterations during development against that of individuals with ASD showed a negative correlation between them ($\rho = -0.48$) (Supplementary Fig. 18b). These data suggest the possibility that PSD composition in individuals with ASD is relatively similar to that in the prenatal or neonatal period compared to healthy subjects. There were 22 DDP-MH and 62 DIP-MH genes, which are upregulated and downregulated in patients with ASD, respectively (Supplementary Fig. 3). They include 18 ASD-related genes registered in Simons Foundation Autism Research Initiative (SFARI), a scientific initiative for understanding, diagnosis, and treatment of ASD (Supplementary Fig. 18c).

## Discussion

In the present study, we revealed the alterations of PSD composition in postnatal mouse and marmoset brains, which occur along with synapse maturation. Systematic bioinformatics analyses revealed possible upstream mechanisms and downstream events of this PSD remodeling. Our study and another study published recently describe the alterations in PSD composition in primate brains during development[78,79]. The proteome datasets obtained here will aid in understanding the molecular mechanisms of synapse alteration in primate brains during postnatal development, especially synapse pruning. Considering the observation of the synapse number in previous reports, synapse pruning may occur more significantly in the postnatal primate brain compared to the mouse brain[3,4,6,7]. The proteins in marmoset PSD whose abundance is significantly altered during postnatal development may include critical protein(s) that regulate synapse pruning in primate brains. It should, however, be noted that the result of the marmoset proteome might be affected by non-synaptic contaminant proteins to some extent because we used crude

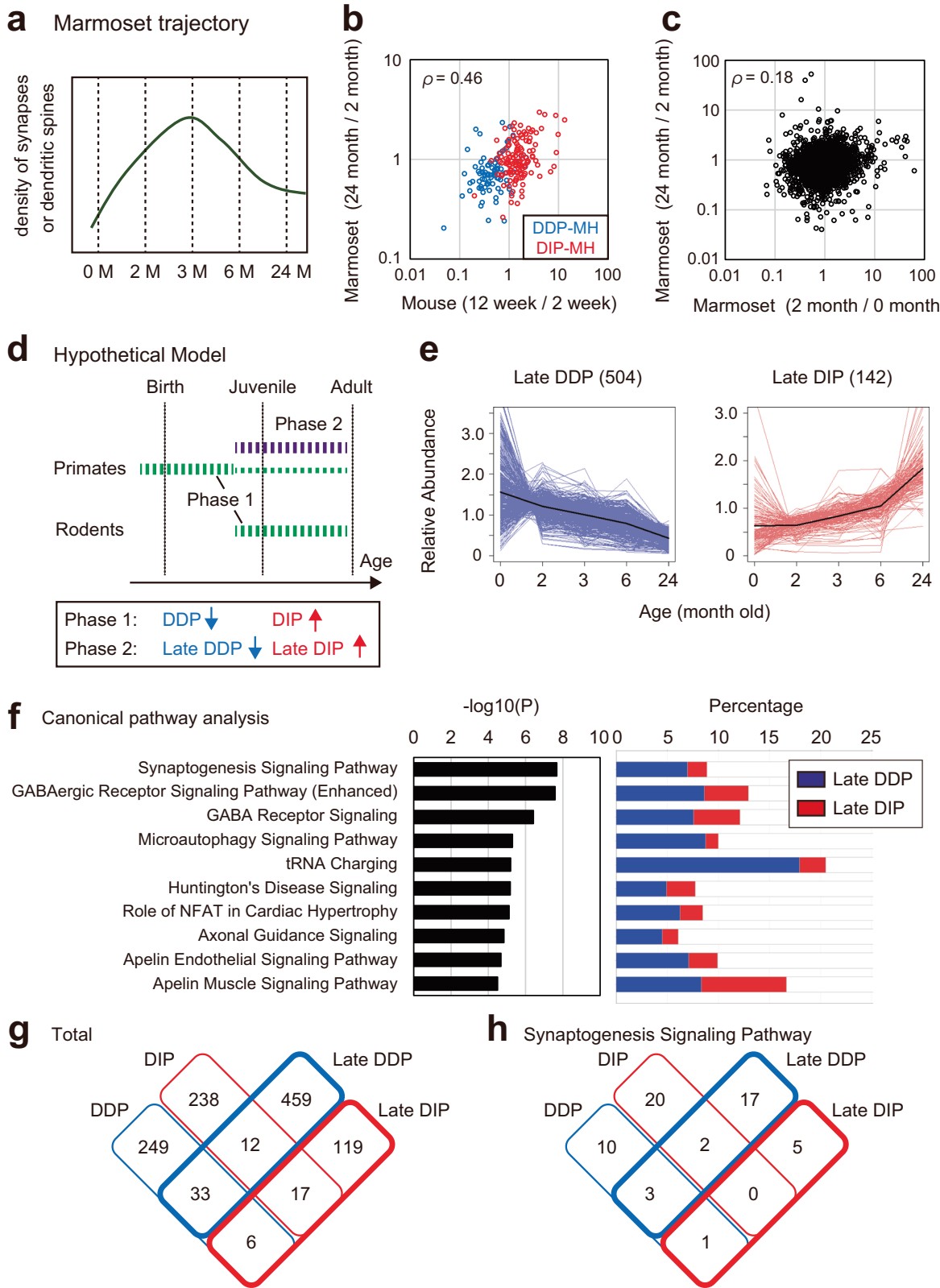

PSD fraction prepared with a 3-step method for marmoset proteome analysis. Another point to note is the absence of marmoset transcriptome data. The remodeling of PSD composition in the marmoset brain within 2 months after birth is presumably preceded by changes in gene expression, similar to what is observed in mice. However, it is currently uncertain whether such gene expression changes occur in the marmoset brain during the perinatal period, as in humans and macaques. The identification of specific transcription regulators upstream of the genes encoding proteins on PSD may explain their regulatory mechanism. A reference to ChIP-seq data showed that genes in proximity to DNA sequences bound to BRD4 are enriched on DNA sequences close to DDP-MH genes. In contrast, they are less enriched on DNA sequences around DIP-MH genes (Supplementary Fig. 9a). BRD4 is a transcription factor involved in neuronal

**Fig. 6 | Alteration of PSD proteome at late developmental stage in marmoset neocortex. a** Schematic diagram of the trajectory of synapse density in marmoset neocortex reported previously[7]. **b** Log10 fold change of the abundance of proteins in DDP-MH and DIP-MH groups in the developmental mouse cortex (12 week-old vs. 2 week-old) plotted against that of marmoset neocortex at neonatal period (12 month-old vs 2 month-old). **c** Log10 fold change of protein abundance in the developmental marmoset neocortex (2-month-old vs. 0 month-old) plotted against that at a later period (24 month-old vs. 2 month-old). **d** Hypothetical model illustrating the trajectory of PSD composition in primate and rodent brains. Phase 1 changes include a decrease of DDP and an increase of DIP and take place at later postnatal development in rodent brain and perinatal-neonatal period in mouse brain. Phase 2 changes include a decrease of Late-DDP and an increase of Late-DIP, taking place at later postnatal development in primate brains. In this period, Phase 1 change also continued to a lower extent. **e** Expression profile of the proteins which showed more than a 2-fold decrease (termed 'late DDP') or increase (termed 'Late DIP') in marmoset brain from 2 month-old to 24 month-old. Black lines indicate the average. **f** Canonical pathway analysis using IPA. Pathways with the top 10 lowest raw *P*-values were shown. (left) Enrichment of indicated terms in DDP and DIP described as *P*-value (right) Percentage of proteins included in each term. **g** and **h** Venn diagram illustrating the overlap of the proteins classified into DDP, DIP, Late DDP, and Late DIP. Total proteins (**g**) or proteins of Synaptogenesis Signaling Pathways (**h**) are described. Source data are provided as a Source Data file.

development and function[80]. On the other hand, genes in proximity to DNA sequences bound to transcriptional repressor REST are enriched only on DNA sequences around DIP-MH genes (Supplementary Fig. 9a). Downregulation of REST expression during the prenatal period (https://hbatlas.org/) may be involved in the upregulation of DIP-MH genes and subsequent maturation of PSD composition. In addition to transcription factors, we found a correlation between the expression of the genes encoding DIP-MH and DDP-MH proteins and their epigenetic status, histone H3K27 acetylation (Supplementary Fig. 9c). Genes that showed increased acetylation after birth of humans are enriched in DIP-MH, whereas that showed decreased acetylation are enriched in DDP-MH in neocortex. Considering that acetylation of H3K27 promotes gene expression in general[81,82], this suggests epigenetic regulation of gene expression that is involved in developmental PSD remodeling.

We also found a potential relationship between PSD composition and ASD. Analysis of transcriptome data of ASD patient brains suggested that PSD composition is "immature" in brains with ASD (Supplementary Fig. 18). Because developmental alteration of PSD composition can affect multiple signaling pathways related to synapse maturation, including Rho family GTPase signaling (Fig. 2d-f), defective maturation of these signals may also be involved in synaptic phenotypes of ASD. Indeed, enhanced Rho family GTPase signaling is observed in Fmr1 KO mice (model of Fragile X syndrome)[83]. A possible mechanism of the altered gene expression in patients with ASD is the disruption of upstream regulators of the genes (Supplementary Fig. 9a, 9b). For example, BRD4, which binds to the promoter sequence of DDP-MH genes, may be involved in ASD. In Fmr1 KO mice, the expression of BRD4 protein is upregulated. Inhibition of BRD4 restores the transcriptional profile and rescues excess dendritic spine formation and abnormal social behaviors in Fmr1 KO mice[84]. Upregulation of DDP-MH genes through BRD4 may cause immaturity of PSD composition and, in turn, cause synaptic and behavioral abnormalities. If this is the case, upstream transcriptional regulators of genes differentially expressed during development can be considered as therapeutic targets for ASD to ameliorate synaptic phenotypes through re-maturation of PSD protein composition.

## Methods

### Ethics statement
Our research complies with all relevant ethical regulations and is approved by RIKEN and Kobe University.

### Animals and tissue preparation
The animal experiments were approved by the Animal Research Committee in RIKEN (Permission number: W2019-2-42 and W2022-2-030) and Kobe University Institutional Animal Care and Use Committee (Permission number: A220510). The housing conditions for the purchased mice are as follows: Temperature: 23–25 C, Humidity: 45–65%, Light-Dark cycle: Light (8:00–20:00) Dark (20:00–8:00). The housing conditions for the marmosets are as follows: Temperature: 27–29 °C, Humidity: 35–75%, Light-Dark cycle: Light (8:00–20:00) Dark (20:00–8:00). Whole-brain samples from 2, 3, 6, and 12-week-old male

ICR mice purchased from Japan SLC Inc. (Shizuoka, Japan) were used. After cervical dislocation, the brain was removed from mice, briefly rinsed with ice-cold HBSS (Hanks' Balanced Salt solution), frozen with liquid nitrogen, and stored at −80 °C before use. PSD purification was performed using whole brains pooled from 24 (2 week-old), 8 (3 week-old), or 4 (6 and 12 week-old) mice, respectively. To obtain PSD-I fraction, 6 (2 week-old) or 3 (12 week-old) brains were used. Note that many young mice should be used due to a low yield of PSD from young mice. For the marmoset brain, we used male common marmosets (*Callithrix jacchus*). We prepared marmosets aged 0, 2, 3, 6, and 24 months, where '0 month-old' refers to newborns or 0 day-old marmosets. Two animals were used for each age group. Additional details can be found in Supplementary Fig. 12a. The animals were sedated with an i.p. injection of ketamine hydrochloride (10 mg/kg) and euthanized via an overdose of sodium pentobarbital (75 mg/kg i.p.). After removing the whole brain, individual brain regions were dissected on ice. The brain samples were frozen with liquid nitrogen and stored at −80 °C before use. To obtain PSD fraction from mice or marmosets using the 3-step method, cortex or cerebellum dissected from 1 animal was used.

### PSD purification
PSDs were prepared from mouse brains using a previously described method with minor modification[20]. All steps were performed at 4 °C or on ice. Homogenization of brains was performed by 12 strokes with Teflon-glass homogenizer in Solution A (0.32 M sucrose, 1 mM NaHCO₃, 1 mM MgCl₂ 0.5 mM CaCl₂, and cOmplete EDTA-free Protease Inhibitor Cocktail). Brain homogenate was centrifuged at 1400 g for 10 min at 4 °C to obtain the pellet and the supernatant fraction. The pellet fraction was resuspended in Solution A with 3 strokes of the homogenizer and centrifuged at 700 g for 10 min at 4 °C. The supernatant of the first and second centrifugation was pooled as an S1 fraction and subjected to the subsequent centrifugation at 13,800 g for 10 min at 4 °C. The resulting pellet was resuspended with Solution B (0.32 M Sucrose and 1 mM NaHCO₃) and 6 strokes of the homogenizer (P2 fraction) and centrifuged in a sucrose density gradient (0.85/1.0/1.2 M) for 2 h at 82,500 g. Purified synaptosomes were collected from the 1.0/1.2 M border and diluted twice with Solution B. The purified synaptosome was lysed by adding an equal volume of solution C (1% TX-100, 0.32 M Sucrose, 12 mM Tris-HCl pH 8.1) and rotation at 4 °C for 15 min. The sample was then centrifuged at 32,800 g for 20 min at 4 °C. The resulting pellet was used as a PSD-I fraction. To obtain the PSD-II fraction, PSD-I was resuspended with Solution B and centrifuged in a sucrose density gradient (1.0/1.5/2.0 M) for 2 h at 201,800 g. The border between 1.0 M and 1.2 M was collected and twice diluted with Solution B. After adding an equal volume of Solution D (1% TX-100, 150 mM KCl), they were centrifuged at 201,800 g for 20 min at 4 °C. The resulting pellet, PSD-II, was stored at −80 °C before LC-MS/MS analysis. For PSD purification from the marmoset brain, a 3-step method was used to maximize the yield[66]. Homogenization of brain samples was performed by 6 strokes with Teflon-glass homogenizer in Solution A' (0.32 M sucrose, 10 mM HEPES (pH 7.4), 2 mM EDTA, 5 mM sodium o-vanadate, 30 mM NaF, and cOmplete EDTA-free Protease Inhibitor Cocktail) (tissue weight: volume ratio is 100 mg:1 ml). Brain

homogenate was centrifuged at 800 g for 15 min at 4 °C the supernatant fraction (S1 fraction). The S1 fraction was then subjected to another centrifugation at 10,000 g for 15 min at 4 °C. The resulting pellet was resuspended with Solution B' (1% Triton X-100, 50 mM HEPES (pH 7.4), 2 mM EDTA, 5 mM EGTA, 5 mM sodium o-vanadate, 30 mM NaF, and cOmplete EDTA-free Protease Inhibitor Cocktail) and subjected to another centrifugation at 30,000 g for 30 min at 4 °C. The resulting pellet was washed once with Solution C' (50 mM HEPES (pH 7.4), 2 mM EDTA, 5 mM EGTA, 5 mM sodium o-vanadate, 30 mM NaF, and cOmplete EDTA-free Protease Inhibitor Cocktail) and stored at −80 °C before LC-MS/MS analysis.

## Immunoblotting

Samples were boiled in sample buffer (62.5 mM Tris-HCl pH6.8, 4% sodium dodecyl sulfate (SDS), 10% glycerol, 0.008% bromophenol blue, and 25 mM DTT). Proteins were separated using SDS-polyacrylamide gel electrophoresis (SDS-PAGE) using 5% (for PLCβ1), 12% (for PALM and β-actin), or 15% (for cofilin and phospho-cofilin) acrylamide gels. Proteins on the gels were transferred onto Immobilon-FL polyvinylidene difluoride membranes (Millipore). The membrane was blocked in blocking buffer (Tris-buffered saline with 0.1% Tween 20 (TBST) and 5% skim milk) at room temperature and then incubated with the indicated primary antibody (1:1000 dilution) overnight in blocking buffer at 4 °C. The membrane was washed three times with TBST, followed by incubation with the respective secondary antibody (1:5000 dilution for fluorophore conjugated antibodies or 1:10000 for peroxidase conjugated antibodies) in blocking buffer for 1 h at room temperature. The membrane was washed five times with TBST. Signal intensities were analyzed using Odyssey near-infrared fluorescence imaging system (LI-COR Biosciences) to detect fluorescence or ImageQuant800 (AMERSHAM) to detect chemiluminescence. Rabbit anti-PSD-95 antibody (ab18258, Abcam), rabbit anti-Synaptophysin antibody (#4329, Cell Signaling Technology), rabbit anti-p-cofilin Ser3 (#3313, Cell Signaling Technology), rabbit anti-cofilin (#5175, Cell Signaling Technology), rabbit anti-Paralemmin-1 antisera[85], rabbit anti-PLCβ1 antibody[86] or mouse anti-β-Actin antibody (A1978, SIGMA) were used as primary antibodies. For fluorescence-based detection, Alexa Fluor 680- conjugated anti-rabbit IgG antibody (A-21076, Thermo Fisher Scientific) or IRDye800CW-conjugated anti-mouse IgG antibody (610-131-121, Rockland Immunochemicals) were used as secondary antibodies. For chemiluminescence detection, peroxidase-conjugated anti-rabbit IgG antibody (111-035-003, Jackson Immuno Research Laboratories, Inc.) or anti-mouse IgG antibody (NA9310, Amersham) were used as secondary antibodies and Immobilon Crescendo (Millipore WBLUR0500) or Chemi-Lumi One Super (Nacalai 02230-14) was used as substrates. The primary antibodies were validated by major band detection at expected molecular weight. See Source Data file for the full scan of immunoblotting.

## Protein analysis by LC-MS/MS

The samples were denatured and reduced with 7 M guanidine-HCl / 1 M Tris (pH8.5) / 10 mM EDTA / 50 mM DTT at 37 °C for 2 h, followed by carboxymethylation with 100 mM sodium iodoacetate at 25 °C for 30 min. The protein was desalted using a PAGE Clean Up Kit (Nacalai tesque) according to the instruction manual. The desalted protein was dissolved using 10 mM Tris-HCl (pH8.0) / 0.03% n-Dodecyl-β-D-maltoside and digested with trypsin (TPCK-treated, Worthington Biochemical) for 12 h. The trypsinized protein fragments were applied to liquid chromatography (LC) (EASY-nLC 1200; Thermo Fisher Scientific, Odense, Denmark) coupled to a Q Exactive HF-X hybrid quadrupole-orbitrap mass spectrometer (Thermo Fisher Scientific, Inc., San Jose, CA, USA) with a nanospray ion source in positive mode. The peptides derived from protein fragments were separated on a NANO-HPLC capillary column C18 (0.075 mm ID x 100 mm length, 3 mm particle size, Nikkyo Technos, Tokyo, Japan). The mobile phase "A" was water

with 0.1% formic acid and the mobile phase "B" was 80% acetonitrile with 0.1% formic acid. A linear gradient was used for 250 min at a flow rate of 300 nL/min: 0%-% B. The Q Exactive HF-X-MS was operating in the top-10 data-dependent scan mode. The parameters of Q Exactive HF-X were as follows: spray voltage, 2.0 kV; capillary temperature, 250 °C; mass range (m/z), 200–2000; normalized collision energy, 27%. Raw data were acquired with Xcalibur software.

## Protein identification

The MS and MS/MS data were searched against the NCBI-nr using Proteome Discoverer version 2.2 (Thermo Fisher Scientific) with the MASCOT search engine software version 2.6 (Matrix Science). The search parameters were as follows: enzyme, trypsin; static modifications, carboxymethyl (Cys); dynamic modifications, acetyl (Protein N-term), Gln- > pyro-Glu (N-term Gln), oxidation (Met); precursor mass tolerance, ±15 ppm; fragment mass tolerance, ±30 mmu; max. missed cleavages, 3. The proteins were identified when their false discovery rates (FDR) were <1%.

## Data analysis and bioinformatics

Data analyses were mainly performed using R software (version 3.4.4 or 4.1.0). Heatmaps were generated in R using the heatmap.2 function in the gplots package. For k-means clustering, the number of clusters was assessed using the NbClust package[87]. Conversion of protein ID (GI number) into gene ID (Entrez Gene ID and official gene symbol) and identification of homologous gene in other species was performed using db2db of bioDBnet[88]. Genes not converted with db2db were individually checked with NCBI Protein (https://www.ncbi.nlm.nih.gov/protein) and NCBI Gene (https://www.ncbi.nlm.nih.gov/gene/). The lists of mouse Entrez Gene ID were used for the following enrichment analyses. Enrichment of proteins reported to be expressed on PSD was evaluated using DAVID version 6.8 (https://david.ncifcrf.gov/home.jsp)[89] with no background list. GO analysis and pathway analysis was performed using Metascape (https://metascape.org)[50] and SynGO (https://www.syngoportal.org/)[51]. For analysis with Metascape, proteins included in previously reported PSD proteome datasets (5412 proteins in 16 datasets) were used as background to avoid overestimating synaptic protein enrichment. *M. musculus* was selected as the Input species and *M. musculus* (for mouse data) or *H. sapiens* (for marmoset data) was chosen as the Analysis species. For other settings, default parameters were used. For analysis with SynGO, brain-expressed proteins were used as background. Enrichment of transcription factor binding on the genes was analyzed using ChIP-Atlas[63]. In Enrichment Analysis of ChIP-Atlas, we input the DDP-MH or DIP-MH list, selected "ChIP: TFs and others" and "Neural" and used "Refseq coding genes" as a background. For other settings, default parameters were used. Enrichment of disease-related genes and transcription binding sites were analyzed using ToppCluster[90]. Canonical pathway analysis and network analysis were performed using Ingenuity Pathway Analysis (IPA) (QIAGEN). In k-means clustering, the optimal number of clusters was determined with the NbClust package of R software. For the mouse PSD proteome dataset, NbClust suggested 2 or 3 clusters as the best number. We adopted 3 clusters, and proteins were classified into three groups as described in the Results.

## Analysis of the expression patterns of genes that encode PSD proteins

Proteome datasets of PSD fraction[34–48] were described in these articles. The full list of proteins is summarized and described in our article[91]. For transcriptome of developing mouse brain, we used two datasets; Dataset 1[30] and Dataset 2[23]. Dataset 1 was downloaded from the NCBI website (https://www.ncbi.nlm.nih.gov//sra/?term=SRP055008) and converted to expression level (TPM) with RSEM[92]. Transcriptome datasets and histone acetylation datasets of the developing human brain[26] and macaque brain[27] were downloaded from the PsychENCODE

website (human transcriptome: http://development.psychencode.org/files/processed_data/RNA-seq/mRNA-seq_hg38.gencode21.wholeGene.geneComposite.STAR.nochrM.gene.RPKM.normalized.CQNCombat.txt) (macaque transcriptome: http://evolution.psychencode.org/files/processed_data/RNA-seq/nhp_development_RPKM_rmTechRep.txt) (human histone acetylation: http://development.psychencode.org/files/processed_data/ChIP-seq/H3K27ac.zip). Genes including zero value were eliminated for the analysis. Alignment of gene list was performed with Entrez Gene ID. To evaluate the change of gene expression before and after birth, mean mRNA abundance after the birth period was divided by that before birth. For transcriptome of human ASD patient brain, data described in[75] was referred. A list of SFARI ASD genes (released on 01-11-2022) was downloaded from the SFARI website (https://gene.sfari.org/database/human-gene/). The ID of proteins or genes in each dataset was converted to Entrez Gene ID using bioDBnet[88] and then the genes encoding the proteins in our proteome dataset were searched. Note that the proteomic and transcriptomic datasets from primates originate from different species, owing to the availability of samples and data.

## Statistics and reproducibility

In the proteome analyses, we extracted major proteins according to the three criteria and eliminated the other proteins from the following analysis unless otherwise stated; (1) at least two unique peptides were identified, (2) quantified in all datasets, and (3) coefficient of variation <100. In the case when multiple proteins are encoded by a single gene, we selected a single protein encoded by a single gene whose signal intensity is highest. Statistical analyses were mostly performed using R software (version 3.4.4 or 4.1.0). One-way analysis of variance (ANOVA) was performed using the oneway.test function. Correction of $p$-value with Benjamini-Hochberg method was performed using the p.adjust function. The correlation coefficient was calculated using the cor.test function. Spearman's rank correlation coefficient was used instead of Pearson correlation coefficient in the case linear correlation is not assumed. Fisher's exact test was performed using the fisher.test function. $T$-test was performed using TTEST function in Microsoft Excel. No statistical method was used to predetermine sample size. We used $n = 4$ for proteome analysis of mouse PSD considering the maximum number of samples that can be subjected manipulated at once and $n = 2$ for proteome analysis of marmoset PSD to confirm reproducibility in the limited sample availability. The experiments were not randomized, and the investigators were not blinded to allocation during experiments and outcome assessment.

## Reporting summary

Further information on research design is available in the Nature Portfolio Reporting Summary linked to this article.

# Data availability

The raw data of proteomics generated in this study have been deposited in the EBI-PRIDE database under accession code PXD048549. The processed proteome data are provided in the Supplementary Data 1, 2, and 5–9. The RNA-seq datasets that were analyzed in this study are described in the previous reports[23,26,27,30,75]. The mouse transcriptomics datasets are found in references [31] and [32], the human brain transcriptomics datasets in references [27] and [78], and the macaque transcriptomic dataset in reference [28]. Other data generated in this study and quantitative data of individual graphs are provided in the Source Data file. Source data are provided with this paper.

# Code availability

All R codes used in this study is available with open access at the GitHub repository (https://github.com/Takeshi-Kaizuka/Kaizuka_Proteomics_2024)[93].

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

## Acknowledgements

We are grateful to RIKEN CBS Research Resources Division for technical help with ultracentrifuge. We thank Kaori Otsuki and Shungo Adachi for the helpful discussion and all the Takumi laboratory technical staff for their assistance in preparing the experimental reagent. T.T. was supported by KAKENHI (16H06316, 16H06463, 21H04813, 23H04233, and 23KK0132), Japan Society of Promotion of Science (JSPS) and Ministry of Education, Culture, Sports, Science, and Technology; Japan Agency for Medical Research and Development, JP21wm0425011; Japan Science and Technology Agency (JPMJMS2299 and JPMJMS229B), Intramural Research Grant (30–9) for Neurological and Psychiatric Disorders of NCNP; the Takeda Science Foundation; Taiju Life Social Welfare Foundation. T.K. was supported by Grant-in-Aid for JSPS Fellows (16J04376) and KAKENHI (18K14830). N.K and H.O. were supported by the Brain/MINDS project of AMED (JP20dm0207001).

## Author contributions

T.K. designed the research, prepared the PSD samples, and performed most data analysis. T.Suzuki and N.D. performed LC-MS/MS and label-free quantification. N.K. performed dissection of marmoset brain with support from T.Shimogori. K.T. helped with sample preparation and mouse dissection. M.W.K. and M.W. contributed antibodies. T.U. provided materials and analyzed Rho family signaling data. T.K. wrote and T.T. revised the manuscript. T.T., N.D. and H.O. supervised the work. All authors commented on the manuscript and approved the final version.

## Competing interests

The authors declare no competing interests.
