## [Peer Review File · Nature Communications]

Remodeling of the postsynaptic proteome in male mice and marmosets during synapse developmentReviewers' Comments:

Reviewer #1 (Remarks to the Author):

The article by Kaizuka et al. entitled "Developmental dynamics of the postsynaptic proteome to understand synaptic maturation and dysmaturation" reports a comprehensive analysis of proteins at the postsynaptic compartment in different mammalian species.

They provide new proteomic data and compare their results with existing datasets. One of the highlights of this study is the characterization of PSD proteins at the juvenile stage which seem to be much less explored compared to the adult stage. This is also a key resource to study synaptic proteins in neurological and psychiatric disorders. Overall, the article is well written and from my expertise the methodologies are correct (for example the choice of the background for protein enrichment seems to be very relevant). Please find below some comments/suggestions.

The introduction is well written, but the last paragraph might need some editing. Especially, it is not always clear what is compared to what. For example, page 6 line 90 "...we found that similar mRNA abundance alteration occurs during a perinatal period in humans and macaques." I presume that it is "alteration" occurring during development, but not certain. The term "alteration" is often used and might lead to some confusion. Since alteration is a "change" or a "difference" compared to a reference, it is not always clear what is the reference especially when it is a change during development. It also confusing when disorders are mentioned (alteration compared to controls or compared to development or both).

There is always missing data in proteomic analyses (proteins not detected in all animals). The authors have chosen to include in their analysis only the proteins quantified in all animals or replicates. Some proteins might not be detectable at 2 weeks and detectable only at 12 weeks. It might be interesting to have a separate list and analysis of these proteins. These proteins might represent technical artifacts but they might also be interesting outliers.

The authors should be acknowledged for their effort to provide extensive datasets useful for new analyses. It would be very helpful to include the protein lists of the 16 datasets used to compare the PSD proteins.

Figure 4 and Supp4: It seems that the heat map is inverted since increases are decreasing... Although the text, color codes for the heat map are very confusing and sometimes I suspect that the direction might be wrong. This is not always the same direction (e.g. Fig 1 Red is -3 Blue +3; Fig 3 Red +3 and Blue -3; Figure 4 Red is -3 Blue +3). Unless there is a reason why I would keep the same color code for the figures (I would choose red for up).

Regarding the ChIP-Seq data analyses, there is some confusion on the enrichment analysis instead of saying "BRD4 is enriched on "Decrease A1" genes, the authors might be more specific and clearly

indicate that “genes at proximity to DNA sequences bound to BRD4” are .. or “predicted BRD4 regulated genes”.

The analysis of the 15q dup mice proteome is interesting but as the authors nicely showed 3 weeks old is maybe not the best period to investigate PSD proteins. An analysis at a later stage might provide more difference between wild-type and mutant mice.

Reviewer #2 (Remarks to the Author):

This study reports a comparative proteomic analysis of PSD composition in the postnatal mouse and marmoset brains during postnatal periods. Here the authors find interesting groups of proteins that are increased or decreased or unchanged at the PSD level and describe their potential functions in synapse remodeling. There are also marmoset brain regional differences that are prominent in the hippocampus and cerebellum. They also report the correlative changes in the PSD compositions of mouse and marmoset postnatal brains, where they find similarities and differences. The authors also explore how PSD composition is altered in the brain of mouse models of ASD using the 15q 11-13 duplication mouse model and find evidence of immature PSD composition in the mutant brain. They also employ various analysis methods and try validation experiments to solidify the conclusions.

Given the lack of information on proteomic changes in the PSD of the different marmoset brain regions during the postnatal period, these data sets provide a valuable resource and insights for future neurobiological studies on marmosets in health and disease.

Major comments:

1. In Figure 7, proteomic data from marmosets younger than 2 months would much strengthen the manuscript because other figures comparing mouse mRNA/protein and human mRNA predict comparable changes in PSD composition around early juvenile and perinatal stages, respectively. This may reveal stronger correlative changes between the increase/decrease groups of PSD in mice and marmoset.
2. The proteomic data on mouse brains (Figure 1) seem to have used whole-brain samples. This might have caused the partial suppression of correlative changes between mice and marmosets. I also wonder whether whole-brain mouse samples and specific brain regions such as the neocortex yields different results in terms of the correlative changes within mice and between mice and marmosets.

Minor comments:

1. Several western blot data support altered protein expression in this manuscript, but it seems to lack generalization. In line 260-261, the authors described "Thus, the developmental changes in mRNA

abundance could be understood as described in Figure 4d.", but this description is based on only VANGL2 and SHANK3 changes in mouse mRNA/protein and human mRNA (Figure 4c). If they can show other proteins in Decrease 1A/Increase 1A (e.g. GRIA1 or NRXN3) are also altered as VANGL2 or SHANK3, respectively, the conclusion above could be strengthened.

2. Figure 7 shows interesting decreases in the PSD levels of ribosome/mitochondria-related proteins. What would be the meaning of these changes? Do mice show similar changes?

3. The authors mention in Discussion the possibility of synapse pruning processes being involved. I wonder if the authors could find pruning-related terms in their GO and pathway analyses such as astrocyte/microglia.

4. I think the use of names such as "Increase", "Increase A", "Increase A1" are confusing. It would be better to use clearer names; this is just a suggestion.

5. What is a criterion for referring to 46 major proteins associated with PSD in Figure 2c? It needs to be clarified.

6. The description in lines 375-377 is confusing because the total classified proteins in 3 clusters are not 1,432, but 1,960 in Figure 6d. Line 377 should read "The 1,960 detected proteins were classified into..."

7. It would be better to understand the overall study if the schematic outline of Figures 6 and 7 (marmoset PSD proteome) is added separately in the Extended Data Figure 2.

8. Figure 3e. It is unclear which paralemmin-1 band is specific.

Reviewer #3 (Remarks to the Author):

The manuscript presents a number of rather independent studies with different levels of interest and technical quality. While a number of interesting observations are being done, in my opinion, the authors do not present data that contributes to importantly increase our understanding of the developmental dynamics of the PSD proteome.

Major Points:

Overall, on the analysis of the proteomic changes occurring in the mouse PSD along development, I do not think the authors have learned much from the molecular characteristics of the PSD in early postnatal development. Which is certainly a relevant and insufficiently investigated topic. It might have been a mistake to only consider for subsequent analysis proteins found in all developmental stages. Those proteins appearing only at 2 or 3 weeks might be particularly relevant to understand the developing PSD. Alternatively, using whole brain might also be confounding, as it is well known that different brain

regions develop at different paces. Thus changes occurring in a particular brain region might be diminished or totally erased by those occurring in other brain regions.

The finding that PSD remodelling is driven by transcriptional changes is interesting and well demonstrated, including the time gap between RNA and protein changes in expression, but not really unexpected. All biological processes are driven by transcriptional changes.

Their result on the species conservation of gene expression changes along development is very interesting (transcriptomics on human, mouse and macaque). And indeed suggests that similar processes of PSD remodelling take place among mammals. Their observation on the different timing of these changes between mice and primates is also interesting. Although it should be placed in the context of the relevant literature, as it is well-known that synaptogenesis starts earlier (embryonically) in humans than in rodents. Their finding that these gene expression changes also occur in other brain regions is also interesting. It is unfortunate that the authors could not get some information on the specific functions performed by the genes/proteins from the “Decreased” or “Increased” groups.

The proteomics study on the ASD model does not identify statistically significant DE proteins. The conclusions raised in this section are tainted by this fact. As above the fact that they performed proteomics on whole brain might be making changes occurring between brain regions. Nevertheless, the finding that PSDs from ASD models might be more “immature” is interesting and might deserve further investigation.

While it is true that some ASD forms are characterised by an increased immaturity of dendritic spines, the opposite (i.e.- increased number of mature mushroom-like, spines), has also been observed. The authors should keep this in consideration in their discussion. It is unlikely that this ‘immaturity’ is common to all ASD types, it might be common to a sub-population of ASDs, and it would be interesting to clarify to which sub-population.

The fact that genes encoding PSD proteins are DE in ASD is interesting, but in line with the well-documented observation that genes expressed at the PSD are particularly relevant to ASD and neurodevelopmental disorders in general.

The number of samples and brain region used in the study with marmoset should be more clearly stated in the text. Only one animal seems to have been used for each of each time point sample. Thus N=1. This is clearly insufficient. The study is determined by the technical variability, which presumably will be very low, as here they find large numbers of proteins being DE after correction for multiple testing. For instance in the adult samples the authors report 1432 DE proteins out of 1960 (> 70%).

Minor Points:

Figure 1a. Data from 6 and 12 weeks do cluster together. Do not draw 2 circles, they are grouped. Text in pages 7 and 8 should be changed accordingly.

Page 8 – line 130-131. It is surprising that the overlap between the data in Ref 47 and that in this manuscript is so low (< 20%). This should be discussed. It might be that Ref 47 data is not of sufficient

quality or not readily comparable, as it has been obtained from a crude PSD preparation. The authors might want a look for better proteomic datasets to compare their proteins to.

It is unclear to this reviewer how this statement “These results suggest that Cluster 1 includes PSD proteins in the juvenile brain, which have not been studied well” should be interpreted. Consider removing or re-writing.

Page 9 – Lane 141: The authors should further develop on this statement: “analysis using SynGO showed that proteins involved in synaptic signalling are enriched in Cluster 1” : what signalling pathways, what proteins...etc. What are Cluster 1 proteins ?

Page 10 – Lane 156-157: Re-write: “We found that key enzymes involved in spine enlargement and synapse stabilization in an “Increase” group....”

Page 10: The fact that the same Ingenuity pathways seem to be enriched in Cluster 1 and Cluster 3 proteins takes interest out of this result. Apparently, the authors could not find what is more characteristic of Cluster 1 proteins (i.e. PSD proteins characteristic of a young PSD) or Cluster 3 proteins (i.e. PSD proteins characteristic of an adult PSD). How is this interpreted by the authors? Can we learn but makes a young PSD different from an adult PSD based on this data?

The authors might want to compare the statistics (i.e. p-value, enrichment fold, etc) provided by Ingenuity from the analysis done by Cluster 1 or Cluster 3 proteins. Are there pathways which are more enriched in one or the other Cluster ?

Are there really no pathways only enriched in one Cluster?

Finally, the output of Ingenuity should be added as a Supplementary Table.

Page 11. Data on p-Cofilin. Provide a graph summarising WB data and add info on the statistics. “We found that phosphorylation of cofilin is decreased upon development in the crude synaptosome fraction obtained from the cortex and cerebellum (Fig. 2f) “ provide statistics to support this claim.

Page 11. There is no need to have this section: “Altered abundance of proteins in inhibitory synapses and electrical synapses”, no conclusion is drawn from the data. Presented in this section.

Pages 16- 17. Brain region used in the proteomics analysis of 15d11-13 mice should be stated here.

Page 17 I disagree with this statement: “Nevertheless, PCA of the proteome data showed a cluster of WT and 15q dup mice and a similar tendency of difference in four biological replicates (Extended Data Fig. 7a).” These groups do not form clear independent clusters.

Lane 321-322: In which cortical region do they focus?

Reviewer #1 (Remarks to the Author):

The article by Kaizuka et al. entitled “Developmental dynamics of the postsynaptic proteome to understand synaptic maturation and dysmaturation” reports a comprehensive analysis of proteins at the postsynaptic compartment in different mammalian species.

They provide new proteomic data and compare their results with existing datasets. One of the highlights of this study is the characterization of PSD proteins at the juvenile stage which seem to be much less explored compared to the adult stage. This is also a key resource to study synaptic proteins in neurological and psychiatric disorders. Overall, the article is well written and from my expertise the methodologies are correct (for example the choice of the background for protein enrichment seems to be very relevant). Please find below some comments/suggestions.

We thank this reviewer for the critical reading and helpful comments. We revised the text as suggested and performed additional experiments to address the concern.

The introduction is well written, but the last paragraph might need some editing. Especially, it is not always clear what is compared to what. For example, page 6 line 90 “...we found that similar mRNA abundance alteration occurs during a perinatal period in humans and macaques.” I presume that it is “alteration” occurring during development, but not certain. The term “alteration” is often used and might lead to some confusion. Since alteration is a “change” or a “difference” compared to a reference, it is not always clear what is the reference especially when it is a change during development. It also confusing when disorders are mentioned (alteration compared to controls or compared to development or both).

We agree with this comment. In the revised text, we avoided use of the words “alteration” or “change” without clarifying the references. The revised text is as follows.

“Systematic bioinformatics analyses using reported transcriptome datasets revealed a positive correlation between the relative abundance of mRNA and proteins on PSD, suggesting that transcriptional regulation is involved in PSD remodeling.”

“Comparing the developmental gene expression changes in human (prenatal vs postnatal) with the transcriptome of ASD patient brains (control vs ASD), we found

that developmental PSD remodeling is thought to be defective in patients with ASD.”

There is always missing data in proteomic analyses (proteins not detected in all animals). The authors have chosen to include in their analysis only the proteins quantified in all animals or replicates. Some proteins might not be detectable at 2 weeks and detectable only at 12 weeks. It might be interesting to have a separate list and analysis of these proteins. These proteins might represent technical artifacts but they might also be interesting outliers.

We agree with this comment. The revised manuscript presented all protein lists detected in developing mouse PSD (Supplementary Table S1). We found 270 proteins that were reproducibly detected at the specific stage but not detected at another specific stage at all (Extended Data Fig. 2a). We further extracted proteins detected mainly in the younger period (Young; 221 proteins) and adult period (Adult; 40 proteins) and analyzed them (Extended Data Fig. 2b-c). As these “stage specific” proteins seem not major components of PSD, we focused on proteins detected in all samples in the following analysis.

The authors should be acknowledged for their effort to provide extensive datasets useful for new analyses. It would be very helpful to include the protein lists of the 16 datasets used to compare the PSD proteins.

Thank you for appreciating our work on this. This is a part of our work summarizing previously published PSD proteome datasets. The full protein list is described in a recently posted Biorxiv paper.

<https://www.biorxiv.org/content/10.1101/2023.01.23.525126v1>

We cited this paper in the revised manuscript.

Figure 4 and Supp4: It seems that the heat map is inverted since increases are decreasing... Although the text, color codes for the heat map are very confusing and sometimes I suspect that the direction might be wrong. This is not always the same direction (e.g. Fig 1 Red is -3 Blue +3; Fig 3 Red +3 and Blue -3; Figure 4 Red is -3 Blue +3). Unless there is a reason why I would keep the same color code for the figures (I would choose red for up).

We thank this reviewer for pointing out this mistake. We modified the color code in the revised manuscript.

Regarding the ChIP-Seq data analyses, there is some confusion on the enrichment analysis instead of saying “BRD4 is enriched on “Decrease A1” genes, the authors might be more specific and clearly indicate that “genes at proximity to DNA sequences bound to BRD4” are .. or “predicted BRD4 regulated genes”.

We modified the text as suggested.

“Reference of ChIP-seq data showed that genes at proximity to DNA sequences bound to BRD4 are enriched on DNA sequences close to DDP-MH genes.”

The analysis of the 15q dup mice proteome is interesting but as the authors nicely showed 3 weeks old is maybe not the best period to investigate PSD proteins. An analysis at a later stage might provide more difference between wild-type and mutant mice.

We thank this reviewer for the critical comments. As suggested, we conducted the proteome analysis of adult (12-week-old)15q dup mice (Supplementary Table S10). We found that the 15q dup mice show more significant differences from wild-type mice at the adult stage (Fig. 7d, 7e, Extended Data Fig. 15a, 15b). The differentially expressed proteins include proteins involved in RhoA signaling (Fig. 7g).

Reviewer #2 (Remarks to the Author):

This study reports a comparative proteomic analysis of PSD composition in the postnatal mouse and marmoset brains during postnatal periods. Here the authors find interesting groups of proteins that are increased or decreased or unchanged at the PSD level and describe their potential functions in synapse remodeling. There are also marmoset brain regional differences that are prominent in the hippocampus and cerebellum. They also report the correlative changes in the PSD compositions of mouse and marmoset postnatal brains, where they find similarities and differences. The authors also explore how PSD composition is altered in the brain of mouse models of ASD using the 15q 11-13 duplication mouse model and find evidence of immature PSD composition in the mutant brain. They also employ various analysis methods and try validation experiments to solidify the conclusions.

Given the lack of information on proteomic changes in the PSD of the different marmoset brain regions during the postnatal period, these data sets provide a valuable resource and insights for future neurobiological studies on marmosets in health and disease.

We thank this reviewer for the critical reading and appreciation of our manuscript. We performed several major experiments and analyses to address the concerns raised here. Of note, we performed proteome analysis of marmoset PSD including neonatal (0 M) sample. This data provides a more comprehensive view of developmental PSD remodeling in primate brains. We also performed proteome analysis of developmental mouse PSD with the same experimental settings to marmoset proteome. This data enabled us to compare mouse and marmoset data more accurately.

Major comments:

1. In Figure 7, proteomic data from marmosets younger than 2 months would much strengthen the manuscript because other figures comparing mouse mRNA/protein and human mRNA predict comparable changes in PSD composition around early juvenile and perinatal stages, respectively. This may reveal stronger correlative changes between the increase/decrease groups of PSD in mice and marmoset.

We thank this reviewer for raising a critical suggestion. We performed proteome analysis of marmoset PSD including neonatal (0 M) marmoset sample. As a result, we found a

positive correlation of PSD remodeling between 2-12 W mouse and 0-2 M marmoset (Fig. 5e). The correlation was especially significant when we focused on “Decrease A1 (DDP-MH)” and “Increase A1 (DIP-MH)” (Fig. 5e and 5f).

2. The proteomic data on mouse brains (Figure 1) seem to have used whole-brain samples. This might have caused the partial suppression of correlative changes between mice and marmosets. I also wonder whether whole-brain mouse samples and specific brain regions such as the neocortex yields different results in terms of the correlative changes within mice and between mice and marmosets.

We again thank this reviewer for raising a critical concern. To address this point, we conducted proteome analysis of a developing mouse brain using the same protocol we used for the marmoset. We analyzed the cortex and cerebellum of mouse PSD prepared with a 3-step method using 2 biological replicates. The significant decrease and increase of Decrease A1 (DDP-MH) and Increase A1 (DIP-MH) are found in both regions. Comparing mouse and marmoset data, we found a positive correlation between PSD remodeling in mice and marmoset (Fig. 5e).

Minor comments:

1. Several western blot data support altered protein expression in this manuscript, but it seems to lack generalization. In line 260-261, the authors described "Thus, the developmental changes in mRNA abundance could be understood as described in Figure 4d.", but this description is based on only VANGL2 and SHANK3 changes in mouse mRNA/protein and human mRNA (Figure 4c). If they can show other proteins in Decrease 1A/Increase 1A (e.g. GRIA1 or NRXN3) are also altered as VANGL2 or SHANK3, respectively, the conclusion above could be strengthened.

We agree with this comment. For generalization, we described a plot of the mean relative abundance of Decrease A1 (DDP-MH) proteins and Increase A1 (DIP-MH) proteins instead of the schematic figure (Fig. 4d). In addition, we also described the data of HNRNPM and CAMK2A in Extended Data Fig. 7 as other representative examples.

2. Figure 7 shows interesting decreases in the PSD levels of ribosome/mitochondria-related proteins. What would be the meaning of these changes? Do mice show similar changes?

In the new data of marmoset proteome, we did not perform cluster analysis for simplification. We extracted proteins that show more than 2-fold changes after 2M (Fig. 6e; Late-DDP and Late-DIP). In those groups, we did not detect pathway terms related to ribosome/mitochondria (Fig. 6f).

3. The authors mention in Discussion the possibility of synapse pruning processes being involved. I wonder if the authors could find pruning-related terms in their GO and pathway analyses such as astrocyte/microglia.

We did not find enrichment of pathways related to astrocyte/microglia in the proteins that showed more than 2-fold changes after 2 M in the marmoset neocortex. We think that part of molecules related to “Synaptogenesis Signaling Pathway” work on synapse pruning. Following sentences are shown in the revised manuscript.

“Pathway analysis using IPA showed that proteins involved in “Synaptogenesis Signaling Pathway” are enriched in Late DDP and Late DIP, suggesting their involvement PSD composition in synapse remodeling at later developmental stages (Fig. 6f, Supplementary Table 3). There is some overlap of pathways between DDP/DIP and Late DDP/DIP, including Synaptogenesis Signaling Pathway, GABAergic Receptor Signaling Pathway (Enhanced), and Huntington’s Disease Signaling (Fig. 2d, 6f). However, there is little overlap of proteins between DDP/DIP and Late DDP/DIP (Fig. 6g). There is also little overlap of proteins classified into Synaptogenesis Signaling Pathway, suggesting that distinct proteins related to synapse regulation are differentially expressed at Phase 1 and Phase 2 (Fig. 6h).”

4. I think the use of names such as “Increase”, “Increase A”, “Increase A1” are confusing. It would be better to use clearer names; this is just a suggestion.

Thanks for the suggestion. In the revised manuscript, we used the following terms instead of them.

Increase -> DIP (developmentally increased postsynaptic proteins)

Increase A -> DIP-M (DIP correlated with mouse transcriptome)

Increase A1 -> DIP-MH (DIP correlated with mouse and human transcriptome)

Decrease -> DDP (developmentally decreased postsynaptic proteins)

Decrease A -> DDP-M (DDP correlated with mouse transcriptome)

Decrease A1 -> DDP-MH (DDP correlated with mouse and human transcriptome)

5. What is a criterion for referring to 46 major proteins associated with PSD in Figure 2c? It needs to be clarified.

We picked up proteins described as major postsynaptic molecules (and their associated molecules) in wide-ranged review articles such as Zhu et al. Nat. Rev. Neurosci. 2016. We cited this paper in the sentence that mentions the 46 major proteins.

6. The description in lines 375-377 is confusing because the total classified proteins in 3 clusters are not 1,432, but 1,960 in Figure 6d. Line 377 should read "The 1,960 detected proteins were classified into..."

We eliminated this sentence as we conducted proteome analysis again. We did not perform cluster analysis on the marmoset data for simplification in the revised manuscript.

7. It would be better to understand the overall study if the schematic outline of Figures 6 and 7 (marmoset PSD proteome) is added separately in the Extended Data Figure 2.

Thanks for the suggestion. We think the problem was too much complexity of marmoset proteome data due to multiple clustering analyses. In the revised manuscript, we avoided unnecessary clustering analyses to simplify the data.

8. Figure 3e. It is unclear which paralemmin-1 band is specific.

All bands are thought to be paralemmin-1 as Paralemmin-1 is known to give multiple bands on Western blots due to differential splicing as well as phosphorylation (Kutzleb et al. Histochem Cell Biol. 2007). We added this information to the figure legend.

Reviewer #3 (Remarks to the Author):

The manuscript presents a number of rather independent studies with different levels of interest and technical quality. While a number of interesting observations are being done, in my opinion, the authors do not present data that contributes to importantly increase our understanding of the developmental dynamics of the PSD proteome.

We thank this reviewer for critical reading of our manuscript. We agree that our previous manuscript shows independent studies as the experimental design of proteome analysis applied for mouse and marmoset differed in various points. To solve this problem, we conducted additional proteome analysis of mouse and marmoset PSD using the same experimental settings. The new results uncover the relationship between mouse PSD and marmoset PSD that was not shown in the previous manuscript. We believe that our revised manuscript contributes to our understanding of the developmental dynamics of the PSD proteome.

Major Points:

Overall, on the analysis of the proteomic changes occurring in the mouse PSD along development, I do not think the authors have learned much from the molecular characteristics of the PSD in early postnatal development. Which is certainly a relevant and insufficiently investigated topic. It might have been a mistake to only consider for subsequent analysis proteins found in all developmental stages. Those proteins appearing only at 2 or 3 weeks might be particularly relevant to understand the developing PSD. Alternatively, using whole brain might also be confounding, as it is well known that different brain regions develop at different paces. Thus changes occurring in a particular brain region might be diminished or totally erased by those occurring in other brain regions.

We agree that proteins appearing only at 2 or 3 weeks might contribute to developing ASD. In the revised manuscript, we first listed all detected proteins (Supplementary Table S1). They include proteins detected only at limited developmental stage(s) (Extended Data Fig. 2a). Those proteins hardly include postsynaptic proteins reported in previous studies (Extended Data Fig. 2b). Instead, proteins that seem unrelated to synapses are enriched in them, such as "spermatogenesis" and "cartilage development involved in endochondral bone morphogenesis" (Extended Data Fig. 2c), suggesting that those

proteins are mainly composed of non-synaptic contaminants. We, therefore, focused on proteins detected in all samples in the following experiments and analyses. As for brain region, we think the whole-brain analysis is well informative considering that age-dependent difference in gene expression and protein expression seems more significant than the region-dependent difference in developing mouse brain (Fertuzinhos et al. Cell Rep 2014, Li et al. Science 2018 Cizeron et al. Science 2021, etc.). Moreover, in the revised manuscript, we showed data of PSD proteome in the cortex and cerebellum in mice and marmoset (Supplementary Table S6-9, Fig. 5, and Extended Data Fig. 12 and 13). Although a more detailed analysis of brain regions may provide further information, it would be a possible future study.

The finding that PSD remodelling is driven by transcriptional changes is interesting and well demonstrated, including the time gap between RNA and protein changes in expression, but not really unexpected. All biological processes are driven by transcriptional changes.

We agree with this comment. Although the correlation between transcriptome and proteome is not unexpected, we provide two important insights in this study. First, we uncovered which protein correlates with transcription and which does not. In some cases, as shown in Fig. 3e, protein abundance in PSD is not correlated with transcription or total protein abundance. Second, we found the time lag between transcriptome changes and proteome changes and raised the idea of explaining this time lag (Fig. 3c, 3d, 4c).

Their result on the species conservation of gene expression changes along development is very interesting (transcriptomics on human, mouse and macaque). And indeed suggests that similar processes of PSD remodelling take place among mammals. Their observation on the different timing of these changes between mice and primates is also interesting. Although it should be placed in the context of the relevant literature, as it is well-known that synaptogenesis starts earlier (embryonically) in humans than in rodents. Their finding that these gene expression changes also occur in other brain regions is also interesting. It is unfortunate that the authors could not get some information on the specific functions performed by the genes/proteins from the “Decreased” or “Increased” groups.

We thank this reviewer for appreciating our study. We think that the differentially expressed proteins are involved in the regulation of synaptic signaling, including the Rho

GTPase pathway (Fig. 2d, 2f). Further analysis of changes in synaptic signaling would be an interesting topic for future research.

The proteomics study on the ASD model does not identify statistically significant DE proteins. The conclusions raised in this section are tainted by this fact. As above the fact that they performed proteomics on whole brain might be making changes occurring between brain regions. Nevertheless, the finding that PSDs from ASD models might be more “immature” is interesting and might deserve further investigation.

While it is true that some ASD forms are characterised by an increased immaturity of dendritic spines, the opposite (i.e.- increased number of mature mushroom-like, spines), has also been observed. The authors should keep this in consideration in their discussion. It is unlikely that this ‘immaturity’ is common to all ASD types, it might be common to a sub-population of ASDs, and it would be interesting to clarify to which sub-population.

We agree that the PSD proteome of the ASD model mouse (15q dup) does not show a significant difference from that of wild-type mice at 3 weeks old. In the revised manuscript, we included data from adult (12-week-old) mice, which showed a more significant difference (Fig. 7d, 7e). However, we found that they do not show a significant tendency of “immaturity” (Fig. 7f). As the reviewer explained, the synaptic phenotype of ASD is not always the same, suggesting the diversity of the ASD pathology. We include this in the Discussion part.

“It should be noted, however, immature PSD composition may not be observed in all cases of ASD, as 15q dup mice did not show a clear tendency of immaturity (Fig. 7f). Considering the diversity of phenotypes observed in ASD (Ref 68), the immaturity explains a subgroup of ASD.”

The fact that genes encoding PSD proteins are DE in ASD is interesting, but in line with the well-documented observation that genes expressed at the PSD are particularly relevant to ASD and neurodevelopmental disorders in general.

We agree with this comment. In this study, we provide new insight into differentially expressed genes encoding PSD proteins, explaining this in the context of “immaturity”.

The number of samples and brain region used in the study with marmoset should be more clearly stated in the text. Only one animal seems to have been used for each of each time

point sample. Thus N=1. This is clearly insufficient. The study is determined by the technical variability, which presumably will be very low, as here they find large numbers of proteins being DE after correction for multiple testing. For instance in the adult samples the authors report 1432 DE proteins out of 1960 (> 70%).

We agree with the insufficiency of sample numbers and the statistical analysis problem using technical replicates. To confirm reproducibility across individuals, we prepared a PSD sample from another marmoset at each age and conducted proteome again. The analysis with N=2 biological replicates described the reproducible tendency of PSD remodeling during postnatal development in the neocortex and cerebellum (Fig. 5c, 5d, and Extended Data Fig. 11b, 11c).

Minor Points:

Figure 1a. Data from 6 and 12 weeks do cluster together. Do not draw 2 circles, they are grouped. Text in pages 7 and 8 should be changed accordingly.

We modified the figure and the related text. Also, we described other panels of PCA using the same method.

Page 8 – line 130-131. It is surprising that the overlap between the data in Ref 47 and that in this manuscript is so low (< 20%). This should be discussed. It might be that Ref 47 data is not of sufficient quality or not readily comparable, as it has been obtained from a crude PSD preparation. The authors might want a look for better proteomic datasets to compare their proteins to.

The overlap rate is not low. The number of proteins detected in Ref 47 (Ref 49 in the revised manuscript) is only 512, and Cluster 1-3 proteins covered 73% of those proteins in total. This point is clarified in the revised text.

“To test this, we referred to the overlap of the Cluster 1-3 proteins with 512 proteins detected in the crude PSD fraction of P9 mouse. We found that 374 (73%) of the proteins in the P9 PSD proteome overlap with the Cluster 1-3 proteins. They consist of 16.8%, 19.7%, and 13.6% of Cluster 1, 2, and 3 proteins, respectively.”

It is unclear to this reviewer how this statement “These results suggest that Cluster 1

includes PSD proteins in the juvenile brain, which have not been studied well” should be interpreted. Consider removing or re-writing.

We revised the description as follows.

“The relatively high overlap rate of Cluster 1 proteins with P9 PSD proteome compared to adult PSD proteome suggest that Cluster 1 includes proteins expressed on PSD specifically in the juvenile brain.”

Page 9 – Lane 141: The authors should further develop on this statement: “analysis using SynGO showed that proteins involved in synaptic signalling are enriched in Cluster 1” : what signalling pathways, what proteins...etc. What are Cluster 1 proteins ?

Proteins related to multiple signaling pathways, such as GSK3 β signaling, Rho GTPase signaling, and MAP kinase signaling, are included in Cluster 1. We described representative molecules as follows.

“Signaling molecules in Cluster 1 include glycogen synthase kinase-3 beta (GSK3B), FERM, ARH/RhoGEF and pleckstrin domain protein 1 (FARP1), mitogen-activated protein kinase 8 (MAPK8), and phospholipase C beta 1 (PLCB1).”

Page 10 – Lane 156-157: Re-write: “We found that key enzymes involved in spine enlargement and synapse stabilization in an “Increase” group....”

We modified the grammatical error as follows.

“We found that key enzymes involved in spine enlargement and synapse stabilization are included in the DIP group, ...”

Page 10: The fact that the same Ingenuity pathways seem to be enriched in Cluster 1 and Cluster 3 proteins takes interest out of this result. Apparently, the authors could not find what is more characteristic of Cluster 1 proteins (i.e. PSD proteins characteristic of a young PSD) or Cluster 3 proteins (i.e. PSD proteins characteristic of an adult PSD). How is this interpreted by the authors? Can we learn but makes a young PSD different from an adult PSD based on this data?

The authors might want a compare the statistics (i.e. p-value, enrichment fold, etc)

provided by Ingenuity from the analysis done by Cluster 1 or Cluster 3 proteins. Are there pathways which are more enriched in one or the other Cluster ?

Are there really no pathways only enriched in one Cluster?

Finally, the output of Ingenuity should be added as a Supplementary Table.

> Apparently, the authors could not find what is more characteristic of Cluster 1 proteins

> How is this interpreted by the authors?

> Can we learn but makes a young PSD different from an adult PSD based on this data?

We could detect critical synaptic pathways when we use only DDP (Cluster 1 DE proteins) or DIP (Cluster 3 DE proteins) as described below (data not shown in the manuscript).

DDP (Cluster 1 DE proteins) only

DIP (Cluster 3 DE proteins) only

In Fig. 2d (figure replaced due to the update of the IPA database), we described enriched pathways in DDP proteins + DIP proteins because we wanted to evaluate the effect of overall changes in protein composition.

> Are there pathways which are more enriched in one or the other Cluster ?

As shown in Fig. 2d, 8 of 10 pathways showed higher overlap with DIP, whereas "Signaling by Rho Family GTPases" and "Remodeling of Epithelial Adherens Junctions" showed higher overlap with DDP.

> Are there really no pathways only enriched in one Cluster?

For example, "Reelin Signaling in Neurons" showed specific enrichment in DDP, whereas "SNARE Signaling Pathway" showed specific enrichment in DIP (see panels above). Although these pathways are potentially interesting, we focused on "Synaptogenesis Signaling Pathway" and "Signaling by Rho Family GTPases" in the text to focus on essential topics.

> Finally, the output of Ingenuity should be added as a Supplementary Table.

We provided the output of the IPA result in Supplementary Table 3

Page 11. Data on p-Cofilin. Provide a graph summarising WB data and add info on the statistics. "We found that phosphorylation of cofilin is decreased upon development in the crude synaptosome fraction obtained from the cortex and cerebellum (Fig. 2f) " provide statistics to support this claim.

As suggested, we performed a statistical analysis of the immunoblotting (Fig. 2f, right). We confirmed that the phosphorylation of cofilin significantly decreased from 2 weeks old to 12 weeks old ($p < 0.05$, Student's t-test).

Page 11. There is no need to have this section: "Altered abundance of proteins in inhibitory synapses and electrical synapses", no conclusion is drawn from the data. Presented in this section.

We removed this section.

Pages 16- 17. Brain region used in the proteomics analysis of 15d11-13 mice should be stated here.

We used the whole brain for this experiment and proteome analysis of a developing mouse brain.

Page 17 I disagree with this statement: “Nevertheless, PCA of the proteome data showed a cluster of WT and 15q dup mice and a similar tendency of difference in four biological replicates (Extended Data Fig. 7a).” These groups do not form clear independent clusters.

We agree that WT and 15q dup mice show relatively small differences at 3 weeks old. In the revised manuscript, we added proteome data from adult mice. In adult mice, a more evident difference was observed between WT and 15q dup (Fig.7d, 7e) compared to 3-week-old (Extended Data Fig.15a, 15b).

Lane 321-322: In which cortical region do they focus?

In the referred paper, they used the medial frontal cortex (Brodmann area 9) and superior temporal gyrus (Brodmann areas 41, 42, or 22).

REVIEWER COMMENTS

Reviewer #2 (Remarks to the Author):

The authors fully addressed my review comments, including two major comments requesting to try proteomic analyses of neonatal marmoset samples and sub-brain regional mouse samples. I do not have additional comments.

Reviewer #3 (Remarks to the Author):

Remarks to authors

The authors present a significantly improved manuscript relative to their previous version. Some inconclusive bits of work have been eliminated, and this results in a somewhat more coherent article. Most of my comments have been addressed and new data has been generated, especially for the primate-related work. A number of interesting findings regarding the dynamics of the PSD proteome along development and between rodents and primates, which were already present in the previous version, have gained strength and interest, finally a number of interesting proteomic sets are provided.

Nevertheless, the manuscript still suffers from being too long (results = 18+ pages), with sections of very variable interest. The manuscript could still benefit from fine-tuning its focus. In my opinion including results of very different relevance seriously damages the main findings in the paper. The following results sections add little or nothing and could be removed or reduced to 1-2 sentences for the discussion:

- Alteration of protein localization to the PSD
- Transcription factors upstream of the genes encoding proteins on PSD
- Dysregulation of PSD maturation in ASD model mice and patients with ASD

Also, the data on PSD brain differences from the marmoset brain, being that it is still done with N=1 (correct ?) could be removed, or greatly reduced to an introductory sentence for the subsequent section.

I was also very much surprised by the space dedicated to ASD and the ASD model in the discussion. Clearly the interest of this manuscript is not there, but in the evo/devo work performed. Finally, I would also recommend the authors to revisit the title [i.e. what is 'synaptic dysmaturation'? I don't think this notion actually exists]. A more focused title, again on the evo/devo discoveries would give a clearer idea of the main findings of the paper.

Major points

1. Proteome analysis of mouse PSD during development

Provide proper MS data in supplementary material. Information provided in Table S1 is insufficient. The data should be organized as in Table S2. This should be applied to any other table presenting MS data. Uniprot IDs should be added to all protein rows in these tables. Important: All MS data should be shared through public repositories (i.e EBI-PRIDE)

In this revised manuscript and in the rebuttal letter the authors reiterate their point that proteins found in their dataset at 2 or 3 weeks of age but not in previous PSD proteomes obtained from ADULT mice/rat/human are not synaptic components and are thus better considered as contaminants. This is unfortunate and an engrained weakness of this part of the study. The authors should remove sentences supporting this idea, which is wrong (i.e. last part of sentence in lane 166).

The fact that GO analysis of proteins specific to week 2 do not retrieve terms related to synaptic function only proves the point that the knowledge in GO about the synapse is based on studies of adult synapses.

2. In the discussion the authors should dwell in the transcriptomic (i.e epigenetic) mechanisms regulating synaptic proteome development. To the best of my knowledge, their findings in this regard could be more novel that they actually realize. I do not think we know to what extend the synaptic proteome is regulated by general transcriptomic events. Many would argue that the synaptic proteome is mostly regulated by neuronal activity. Disentangling which (may be functional ?) modules of the synaptic proteome are transcriptomically regulated and which not I do not think is something we understand.

3. The authors could also discuss their data in the context of the notion of neoteny. That is a protracted brain development characteristic of humans. The fact that synapses start to appear sooner in primates than rodents, which is well established, sort of goes against the notion of neoteny. It might be that synaptic formation start before that of rodents in humans (as also shown in this manuscript), but also finishes way much later.

Minor points

Lane 116. 'Adult', define; 12 weeks?

Lane 145. Sentence starting by 'The relative...' can be removed, is wrong. The difference between 16.8% (cluster 1) and 13.6% (Cluster 3) does not seem to be much relevant.

Lane 155. 'Considering the term "behaviour", they may...' Please rewrite this sentence, unclear.

Lane 162 to 165. First the authors state that proteins were detected ONLY in specific stages; next the refer to detected MAINLY or detected PRIMARILY. Which hone ot is? Is it only or mostly? Please clarify.

Lane 195. Consider removing the sentence starting by 'This suggest that...', as the authors run the risk to state the obvious.

Lane 216. I do not think it is a good idea to emphasize too much that there is a good correlation between RNA and protein abundance. There will be many more articles showing the opposite.

Lane 257. State the region in which transcriptomics was done (cortex)

Lane 340. Clearly state the age(s) at which the samples were taken

Lane 413. It is not new that that PSD proteins include many genes associated to ASD. This sentence could be removed or the relevant literature could be cited.

Reviewer #4 (Remarks to the Author):

This study uses proteomics on PSD samples as the basis to interrogate the development of synapses in mammalian models across development and adult stages, using mice and marmosets as models. Overall I agree with some of the Reviewer comments that this study looked like multiple parallel studies and it lacked some cohesion. The work done is overall interesting and important, and the revisions did improve the study with additional analyses. However, there are some important questions that remain:

-What is the difference/similarity for marmosets 0-2 months of age versus more than 2 weeks in mice? This comparison is used in the manuscript, but one of the claims is that the PSD changes seen in mice after 2 weeks (postnatal) is observed earlier in marmosets (perinatal) – for the non expert, what other neurodevelopmental milestones are happening in mice vs marmosets?

-There is no major analysis done on excitatory/inhibitory proteins/synapses across both species? I would figure this would be important. I know you cannot measure synapses using PSD proteomics, but the developmental dynamics of Excitatory and inhibitory synapse proteins is important as a comparison between mice/marmosets. More analysis of the species-specific regulation is needed to make the data impactful

-the major issue I have with this manuscript is that if this is the first dataset of marmosets for PSD analysis across time – this is interesting but needs more analysis as a comparison to mouse, to showcase differences between species (see above)

-Further, the final data on the 15q DUP story on PSD is interesting but seems parallel and unrelated to the Marmoset data (primate data) – I don't know how adding an ASD angle only in mice relates to the marmoset/species specific differences/similarities data? If the ASD model was in marmosets then I would way the data would be interesting - but there have already been previous ASD Models studied using proteomics and PSD proteomics, so this is not novel

-Technical note: the comparison of mouse cortex transcriptomics to PSD from whole mouse brains is not very comparable

-Importantly, there is no functional data to test role of marmoset PSD proteins – Arnold Kriegstein published a paper earlier this year on human brain PSD dynamics, and also had some functional data (ie testing a hypothesis from the omics data using live neuron cultures/organoids, etc) – further, they should be citing the Kriegstein proteomics paper

REVIEWER COMMENTS

Reviewer #2 (Remarks to the Author):

The authors fully addressed my review comments, including two major comments requesting to try proteomic analyses of neonatal marmoset samples and sub-brain regional mouse samples. I do not have additional comments.

Many thanks

Reviewer #3 (Remarks to the Author):

Remarks to authors

The authors present a significantly improved manuscript relative to their previous version. Some inconclusive bits of work have been eliminated, and this results in a somewhat more coherent article. Most of my comments have been addressed and new data has been generated, especially for the primate-related work. A number of interesting findings regarding the dynamics of the PSD proteome along development and between rodents and primates, which were already present in the previous version, have gained strength and interest, finally a number of interesting proteomic sets are provided.

We thank this reviewer for critical reading and appreciation of our manuscript.

Nevertheless, the manuscript still suffers from being too long (results = 18+ pages), with sections of very variable interest. The manuscript could still benefit from fine-tuning its focus. In my opinion including results of very different relevance seriously damages the main findings in the paper. The following results sections add little or nothing and could be removed or reduced to 1-2 sentences for the discussion:

- Alteration of protein localization to the PSD
- Transcription factors upstream of the genes encoding proteins on PSD
- Dysregulation of PSD maturation in ASD model mice and patients with ASD

We agree with that and modified the manuscript as follows.

>Alteration of protein localization to the PSD

We moved Fig. 3e to Supplementary Fig. 6 and reduced the text to 2 sentences.

>Transcription factors upstream of the genes encoding proteins on PSD

We removed Fig. 4e and reduced the main text regarding transcriptional factors to 2 sentences.

>Dysregulation of PSD maturation in ASD model mice and patients with ASD

We removed the data and text regarding ASD model mice (15q Dup). Although the data on patients with ASD is still in the text, we reduced the text in this section to about 40% (879 words > 367 words).

Also, the data on PSD brain differences from the marmoset brain, being that it is still done with N=1 (correct ?) could be removed, or greatly reduced to an introductory sentence for the subsequent section.

We moved the data to supplementary figures (Supplementary Fig. 12b-c) and simplified the text.

I was also very much surprised by the space dedicated to ASD and the ASD model in the discussion. Clearly the interest of this manuscript is not there, but in the evo/devo work performed. Finally, I would also recommend the authors to revisit the title [i.e. what is 'synaptic dysmaturation'? I don't think this notion actually exists]. A more focused title, again on the evo/devo discoveries would give a clearer idea of the main findings of the paper.

We agree that ASD is no longer a major issue in this manuscript. We shortened the text related to ASD in the main text and discussion. In the Discussion part, we reduced the text regarding ASD to about 40% (562 words > 215 words). Also, we changed the title, removing the term "dysmatulation".

Major points

1. Proteome analysis of mouse PSD during development

Provide proper MS data in supplementary material. Information provided in Table S1 is insufficient. The data should be organized as in Table S2. This should be applied to any other table presenting MS data. Uniprot IDs should be added to all protein rows in these tables. Important: All MS data should be shared through public repositories (i.e EBI-PRIDE)

We added the information to Table S1 and included Uniprot ID in all supplementary tables. Also, we will share all MS data described in this paper through the public repository, EBI-PRIDE.

In this revised manuscript and in the rebuttal letter the authors reiterate their point that proteins found in their dataset at 2 or 3 weeks of age but not in previous PSD proteomes obtained from ADULT mice/rat/human are not synaptic components and are thus better considered as contaminants. This is unfortunate and an engrained weakness of this part of the study. The authors should remove sentences supporting this idea, which is wrong (i.e. last part of sentence in lane 166).

The fact that GO analysis of proteins specific to week 2 do not retrieve terms related to synaptic function only proves the point that the knowledge in GO about the synapse is based on studies of adult synapses.

We agree with this comment and rewrote the texts as described below.

The particularly low overlap of “Young” group proteins with known PSD proteins suggests that they are proteins expressed on PSD only at a young age and have been overlooked so far.

2. In the discussion the authors should dwell in the transcriptomic (i.e epigenetic) mechanisms regulating synaptic proteome development. To the best of my knowledge, their findings in this regard could be more novel that they actually realize. I do not think we know to what extent the synaptic proteome is regulated by general transcriptomic events. Many would argue that the synaptic proteome is mostly regulated by neuronal activity. Disentangling which (may be functional ?) modules of the synaptic proteome are transcriptomically regulated and which not I do not think is something we understand.

We appreciate this comment. In the revised manuscript, we discussed the epigenetic regulation of PSD genes. For this, we also added the data that show the correlation between epigenetic status and synaptic gene expression (Supplementary Fig. 9c).

3. The authors could also discuss their data in the context of the notion of neoteny. That is a protracted brain development characteristic of humans. The fact that synapses start to appear sooner in primates than rodents, which is well established, sort of goes against the notion of neoteny. It might be that synaptic formation start before that of rodents in

humans (as also shown in this manuscript), but also finishes way much later.

We appreciate this comment. Neoteny is a human-specific feature distinct from other primates (Somel et al. PNAS. 106:5743-8. 2009, Tottenham. Biol Psychiatry. 2020 87:350-358). Although we showed the difference between rodents and primates, we don't have any data that shows the difference between human and non-human primates. For this reason, we decided not to discuss this issue in this manuscript.

Minor points

Lane 116. 'Adult', define; 12 weeks?

Yes. We modified the text.

Lane 145. Sentence starting by 'The relative...' can be removed, is wrong. The difference between 16.8% (cluster 1) and 13.6% (Cluster 3) does not seem to be much relevant.

We removed that phrase and simply mentioned Cluster 1 together with Cluster 2/3, as described below.

>They consist of 16.8%, 19.7%, and 13.6% of Cluster 1, 2, and 3 proteins, respectively, suggesting that. The relatively high overlap rate of Cluster 1 proteins with the P9 PSD proteome compared to adult PSD proteome suggests that Cluster 1 includes proteins expressed on PSD specifically in the juvenile brain, as well as Cluster 2 and Cluster 3.

Lane 155. 'Considering the term "behaviour", they may...' Please rewrite this sentence, unclear.

We clarified this sentence as follows.

> Considering the term "behavior", Cluster 3 proteins may also be involved in the alteration of behavioral properties, such as increased activity during P28-42 and decreased fearful behavior during P24-75

Lane 162 to 165. First the authors state that proteins were detected ONLY in specific

stages; next the refer to detected MAINLY or detected PRIMARILY. Which one of is? Is it only or mostly? Please clarify.

We rewrote the text to clarify the definition of each word, as shown below.

>Among them, we found 270 stage-specific proteins (proteins detected in 4 of 4 replicates and 0 of 4 replicates in at least one stage, respectively) (Supplementary Data 1).

>We also analyzed the 270 stage-specific proteins. Two hundred twenty-one proteins detected in 4 of 4 replicates at 2- and/or 3-week-old are grouped as young age-specific proteins (termed “Young”), and 40 proteins detected in 4 of 4 replicates at 6- and/or 12-week-old are grouped as adult-specific proteins (termed “Adult”).

Lane 195. Consider removing the sentence starting by ‘This suggest that...’, as the authors run the risk to state the obvious.

We understand this concern. To avoid stating the obvious, we replaced the phrase “DE proteins on PSD” with “differential expression of the proteins on PSD”.

>This suggests that differential expression of the proteins on PSD is involved in the alteration of synaptic properties across development.

Lane 216. I do not think it is a good idea to emphasize too much that there is a good correlation between RNA and protein abundance. There will be many more articles showing the opposite.

We agree with this comment. The sentence has been removed.

Lane 257. State the region in which transcriptomics was done (cortex)

We added the information.

Lane 340. Clearly state the age(s) at which the samples were taken

We clarified the age as follows.

>PSD samples were prepared from two biological replicates of the developing mouse cortex and cerebellum (at 2, 3, 6, or 12-week-old) with the 3-step method.

Lane 413. It is not new that that PSD proteins include many genes associated to ASD. This sentence could be removed or the relevant literature could be cited.

We rewrote the sentences as follows.

>It is reported that genes encoding synaptic proteins are enriched in DE genes in ASD patient brains 74,75. Consistent with this, the genes encoding proteins that we detected on PSD (2,186 proteins in Supplementary Table 2) are enriched in DE genes of ASD patients (Fig. 7a).

Reviewer #4 (Remarks to the Author):

This study uses proteomics on PSD samples as the basis to interrogate the development of synapses in mammalian models across development and adult stages, using mice and marmosets as models. Overall I agree with some of the Reviewer comments that this study looked like multiple parallel studies and it lacked some cohesion. The work done is overall interesting and important, and the revisions did improve the study with additional analyses. However, there are some important questions that remain:

We thank this reviewer for critical reading and appreciation of our manuscript.

-What is the difference/similarity for marmosets 0-2 months of age versus more than 2 weeks in mice? This comparison is used in the manuscript, but one of the claims is that the PSD changes seen in mice after 2 weeks (postnatal) is observed earlier in marmosets (perinatal) – for the non expert, what other neurodevelopmental milestones are happening in mice vs marmosets?

We included a more detailed introduction about mouse and marmoset development in the revised manuscript citing two papers.

Considering the degree of cortical laminae maturation, level of serotonin, and expression of genes including serotonin receptor, P0 marmoset is assumed to be the equivalent stage of the mouse at 2~3 weeks old 67,68 This suggests that alteration of PSD composition in mouse takes place in marmoset brain during neonatal period, as well as human and macaque.

-There is no major analysis done on excitatory/inhibitory proteins/synapses across both species? I would figure this would be important. I know you cannot measure synapses using PSD proteomics, but the developmental dynamics of Excitatory and inhibitory synapse proteins is important as a comparison between mice/marmosets. More analysis of the species-specific regulation is needed to make the data impactful

In the revised manuscript, we discuss the excitatory/inhibitory synapses data, comparing the protein abundance of GABAA receptor subunits and ionotropic glutamate receptor subunits. As a result, we found a high abundance and further developmental increase of

the GABAA receptor subunit in the marmoset brain (Supplementary Fig. 16).

-the major issue I have with this manuscript is that if this is the first dataset of marmosets for PSD analysis across time – this is interesting but needs more analysis as a comparison to mouse, to showcase differences between species (see above)

To show the difference between species, we analyzed the correlation of protein abundance between mice and marmosets based on LFQ intensity (Fig. 5c and S15c). This analysis revealed that the protein composition of PSD in mouse cortex at 2-week-old and 12-week-old is similar to marmoset neocortex at 0-month-old and 2~3-month-old, respectively.

-Further, the final data on the 15q DUP story on PSD is interesting but seems parallel and unrelated to the Marmoset data (primate data) – I don't know how adding an ASD angle only in mice relates to the marmoset/species specific differences/similarities data? If the ASD model was in marmosets then I would way the data would be interesting - but there have already been previous ASD Models studied using proteomics and PSD proteomics, so this is not novel

We agree that the 15q Dup story is parallel and unrelated to the other part of this manuscript. In the revised manuscript, we removed all the data and text about the 15q dup mouse.

-Technical note: the comparison of mouse cortex transcriptomics to PSD from whole mouse brains is not very comparable

We understand this concern. We assume this is not a major problem because our whole-brain PSD proteome reflects mainly PSD composition in the cortex, considering the relatively low volume and low PSD yield of the hindbrain.

-Importantly, there is no functional data to test role of marmoset PSD proteins – Arnold Kriegstein published a paper earlier this year on human brain PSD dynamics, and also had some functional data (ie testing a hypothesis from the omics data using live neuron

cultures/organoids, etc) – further, they should be citing the Kriegstein proteomics paper

As suggested by the editor, we decided not to add experimental data about this. The functional role of marmoset PSD proteins would be an interesting issue for future studies. We cited the Kriegstein proteomics paper in the revised manuscript.

REVIEWERS' COMMENTS

Reviewer #3 (Remarks to the Author):

The authors appropriately addressed all my concerns. Thus, I do not have additional remarks.

Reviewer #4 (Remarks to the Author):

The authors have answered all of my questions/comments. Thank you.